# Annual evolution of the ice–ocean interaction beneath landfast ice in Prydz Bay, East Antarctica

Haihan Hu[1], Jiechen Zhao[2, 3*], Petra Heil[4], Zhiliang Qin[2, 3], Jingkai Ma[5], Fengming Hui[1*], Xiao Cheng[1]

[1]School of Geospatial Engineering and Science, Sun Yat-sen University, and Southern Marine Science and Engineering Guangdong Laboratory (Zhuhai), Zhuhai 519082, China;
[2]Qingdao Innovation and Development Base (Centre) of Harbin Engineering University, Qingdao, 266500, China;
[3]College of Underwater Acoustic Engineering, Harbin Engineering University, Harbin 150001, China;
[4]Australia Antarctic Division & Australian Antarctic Programmer Partnership, Private Bag 80, Hobart TAS 7001, Australia;
[5]Key Laboratory of Research on Marine Hazards Forecasting, National Marine Environmental Forecasting Centre, Beijing 100081, China.

*Correspondence to*: Jiechen Zhao (zhaojiechen@hrbeu.edu.cn) and Fengming Hui (huifm@mail.sysu.edu.cn)

**Abstract:** High-frequency observations of the ice–ocean interaction and high-precision estimation of the ice–ocean heat exchange are critical to understanding the thermodynamics of the landfast ice mass balance in Antarctica. To investigate the oceanic contribution to the evolution of the landfast ice, an integrated ocean observation system, including an acoustic Doppler velocimeter (ADV), conductivity–temperature–depth (CTD) sensors, and a sea ice mass balance array (SIMBA), was deployed on the landfast ice near Chinese Zhongshan Station in Prydz Bay, East Antarctica from April to November 2021. The CTD sensors recorded the ocean temperature and salinity. The ocean temperature experienced a rapid increase in late April, from −1.62°C to the maximum of −1.30°C, and then, it gradually decreased to −1.75°C in May and remained at this temperature until November. The seawater salinity and density exhibited similar increasing trends during April and May, with mean rates of 0.04 psu day$^{-1}$ and 0.03 kg m$^{-3}$ day$^{-1}$, respectively, which was related to the strong salt rejection caused by freezing of the landfast ice. The ocean current observed by the ADV had annual mean horizontal and vertical velocities of 9.5±3.9 cm s$^{-1}$ and 0.2±0.8 cm s$^{-1}$, respectively. The domain current direction was SEE (120°)–SWW (240°), and the domain velocity (79%) was 5–15 cm s$^{-1}$. The oceanic heat flux ($F_w$) estimated using the residual method reached a peak of 41.3±9.8 W m$^{-2}$ in April, and then, it gradually decreased to a stable level of 7.8±2.9 W m$^{-2}$ from June to October. The $F_w$ values calculated using three different bulk parameterizations exhibited similar trends with different magnitudes due to the uncertainties of the empirical friction velocity. The spectral analysis results suggest that all of the observed ocean variables exhibited a typical half-day period, indicating the strong diurnal influence of the local tidal oscillations. The large-scale sea ice distribution and ocean circulation contributed to the seasonal variations in the ocean variables, revealing the important relationship between the large-scale and local phenomena. The high frequency and long-term observations of oceanic variables obtained in this study allow us to deeply investigate their diurnal and seasonal variations and to evaluate their influences on the landfast ice evolution.

## 1 Introduction

Since the late 19th century, the Earth has warmed by 0.8°C, while the Arctic warmed by 2–3°C during the same period (Overland et al., 2014). In contrast, there has been no significant warming trend in Antarctica in the past 20 years (Post et al., 2019). Antarctic sea ice plays a critical role in driving and modulating global climate change and local marine and ecosystem systems (Massom and Stammerjohn, 2010). However, in contrast to the rapid decline of the sea ice extent in the Arctic, the Antarctic has experienced a slight increase since the late 1970s (Comiso et al., 2008; Liu and Curry, 2010), with an extended peak of 20 million $km^2$ observed in 2014, after which the summer minima and winter maxima exhibited decreasing trend until they reached a new low in 2021/22 (Parkinson and DiGirolamo, 2021; Raphael and Handcock, 2022; Wang et al., 2022).

Landfast ice commonly exists along the Antarctic coast and is usually attached to the shorelines, ice shelves, glacier tongues, grounded icebergs, or shoals (Massom et al., 2001; Li et al., 2020; Fraser et al., 2021). In contrast to pack ice floes, landfast ice generally has a longer annual duration and a larger thickness, and its width can reach tens to hundreds of kilometres from the shore (Fraser et al., 2021). In winter in the Southern Hemisphere, landfast ice accounts for 3–4% of the total sea ice area (Li et al., 2020) and a larger percentage, approximately 14–20%, of the total sea ice volume (Fedotov et al., 2013). In particular, the proportion of landfast ice off the coast of East Antarctica is larger than that in other Antarctic regions (Giles et al., 2008; Li et al., 2020). As a natural boundary between the ocean and atmosphere, landfast ice strongly influences air–ocean interactions and heat and momentum exchange (Maykut and Untersteiner, 1971; Heil et al., 1996; Heil, 2006). The existence of landfast ice provides an efficient barrier to glaciers and ice sheets, preventing them from calving and vanishing into the Southern Ocean (Massom and Stammerjohn, 2010; Miles et al., 2017).

The growth of landfast ice is mainly attributed to thermodynamic processes. The oceanic heat flux plays a critical role in the ice mass balance and influences the annual growth of landfast ice (Parkinson and Washington, 1979). The main challenge in studying ice–sea heat exchange is developing a method for accurately quantifying the oceanic heat flux and its seasonal variations. However, the oceanic heat flux is difficult to observe directly and is usually estimated by measuring the ice temperature and thickness, known as the residual energy method (McPhee and Untersteiner, 1982). Heil et al. (1996) estimated the annual oceanic heat flux to be 5–12 W $m^{-2}$ based on ice observations at Australia's Antarctica Mawson Station. Lei et al. (2010) studied the seasonal variations in landfast ice in Prydz Bay in 2006 and obtained an oceanic heat flux of 11.8±3.5 W $m^{-2}$ in April and an annual minimum of 1.9±2.4 W $m^{-2}$ in September based on the residual method. Yang et al. (2016) analysed the oceanic heat flux in Prydz Bay using the high-resolution thermodynamic snow and ice (HIGHTSI) model (Launiainen and Cheng, 1998; Vihma, 2002; Cheng et al., 2006) and concluded that it gradually decreased from 25 W $m^{-2}$ to 5 W $m^{-2}$ in winter. Zhao et al. (2019) estimated the oceanic heat flux using the residual method and found that the monthly oceanic heat flux in 2012 was 30 W $m^{-2}$ in March–May, decreased to 10 W $m^{-2}$ during July–October, and increased back to 15 W $m^{-2}$ in November. In terms of the evolution mechanism of the oceanic heat flux, Allison (1981) found that the oceanic heat flux under the landfast

ice near Mawson Station exhibited two peaks throughout the season due to the influence of the thermohaline convection caused
by salt rejection and seasonal variations in the large-scale meridional thermal advection in the Southern Ocean. McPhee et al.
(1996) found that the oceanic heat flux changed on the sub-diurnal scale due to the sub-glacial cold and warm currents. High-
frequency processes such as ocean tides and salt flux have an hourly impact on the oceanic heat flux, making it difficult for
the residual method to capture short-term changes (Lei et al., 2010). Another more accurate approach to estimating the oceanic
heat flux involves direct measurements of the turbulent vertical velocity and high-frequency temperature fluctuations or
measurements of the frictional velocity and temperature difference from the ice–ocean interface to the mixed layer. However,
this method requires precise and high-frequency measurements of the ocean current under the ice and the mixed layer
temperature. This method has been widely used in previous studies conducted in the Arctic and Antarctic (McPhee, 1992;
McPhee et al., 1996, 2008; Maykut and McPhee, 1995; Sirevaag, 2009; Sirevaag and Fer, 2009; Kirillov et al., 2015; Peterson
et al., 2017; Lei et al., 2022). Nonetheless, there is a lack of such detailed and high-frequency landfast ice–ocean observation
data for Prydz Bay, Antarctica.

Direct observations of high-frequency ocean temperature, salinity, and velocity beneath landfast ice are important for filling
the data gap of the ice–ocean interaction near the Chinese Antarctic Zhongshan Station and for more accurately understanding
how the oceanic heat flux affects the growth of sea ice in Prydz Bay on the diurnal and seasonal scales. In this study, a set of
ice–ocean equipment, including an acoustic Doppler velocimeter (ADV), conductivity–temperature–depth (CTD) sensors, and
a sea ice mass balance array (SIMBA), was deployed at a landfast ice site located approximately 1 km far from Zhongshan
Station during April–November 2021. The details of the field observations are presented in Section 2. The observations were
deeply analysed and the oceanic heat flux was estimated using two different methods, i.e., the residual method and the bulk
parameterization method, which are described in Section 3. The relationship between the tides and the oceanic heat flux, as
well as the large-scale and local phenomena, are discussed in Section 4. The conclusions are presented in Section 5.
**2 Data and Methods**
**2.1 Field observations**
The field observations were conducted at Zhongshan Station (69°22′ S, 76°22′ E), the second Chinese Antarctic scientific
research station, which was established in February 1989 and has been operated year-round since its establishment. Zhongshan
Station is located in Prydz Bay, East Antarctica (Fig. 1a), and is surrounded by a 40–100 km wide section of landfast ice in
the cold season, from February to December (Zhao et al., 2020). In the austral summer (i.e., late January), the landfast ice
usually breaks into small floes due to mechanical forcings such as wind, waves, and tides, and then, it completely disappears
(Li et al., 2020), with the exception of some small ice floes in the narrow fjords that survive to become second or multi-year
sea ice in the subsequent winter.

From April 16 to November 7, 2021, an integrated ice–ocean interaction observation system was established by the wintering
team at the coastal landfast ice site, approximately 1 km from Zhongshan Station (Fig. 1b). A cable-type CTD sensor (model:
ALEC ACTD–DF, Japanese JFE Advantech Co., Ltd.) (for more information, see
https://www.xylem.com/siteassets/brand/sontek/resources/specification/sontek-argonaut-adv-brochure-s11-02-1119.pdf, last
access: February 24, 2023) was deployed 2 m beneath the ice surface and 15 m from the shoreline. The CTD measured the
ocean conductivity, temperature, and depth at a frequency of 30 s, with accuracies of $\pm0.02$ mS cm$^{-1}$ ($\pm0.03$ psu) for
conductivity (salinity) and $\pm0.02$°C for temperature. An ADV (model: SonTek Argonaut–ADV, the xylem company) (for more
information, see https://www.analyticalsolns.com.au/product/conductivity_temperature_depth_logger_miniature_.html, last
access: February 24, 2023) was deployed to observe the 3-D ocean velocity at 5 m below the ice surface and 5 m north of the
CTD. The frequency of the ocean velocity observations was 40 s, and the accuracy was $\pm0.001$ m s$^{-1}$. A SIMBA (model: SRSL
SIMBA) (for more information, see https://www.sams-enterprise.com/services/autonomous-ice-measurement/, last access:
February 24, 2023) was deployed 5 m north of the ADV, which contained 240 temperature sensors at 2-cm intervals mounted
on a thermistor string. The 4.8 m long SIMBA temperature chains recorded the vertical temperature profiles of the air–snow–
ice–ocean every 6 hours. The SIMBA had a resolution of $\pm0.0625$°C. The water depths at the CTD, ADV, and SIMBA sites
were 4.5 m, 13 m, and 13 m, respectively. Manual observations, including snow and ice thickness measurements, were
conducted around the integrated ice–ocean interaction observation system every five days by the wintering team.

Due to the effect of the extremely cold conditions on the battery power supply, the observation system stopped working during
part of the period, April 24–May 11 for the ADV and July 7–15 for the CTD. A data quality control was applied to the original
time series to pick out the anomalous values. To match the different frequencies of the ADV and CTD in the inter-comparisons
and the analysis of the oceanic heat flux, the observations were averaged and integrated into a new time series with 2-minute
intervals. Regarding the processing of the SIMBA observation data, 3-point smoothing was introduced to minimize the noise
influences, which has been used by Zhao et al. (2017) and Tian et al. (2017).

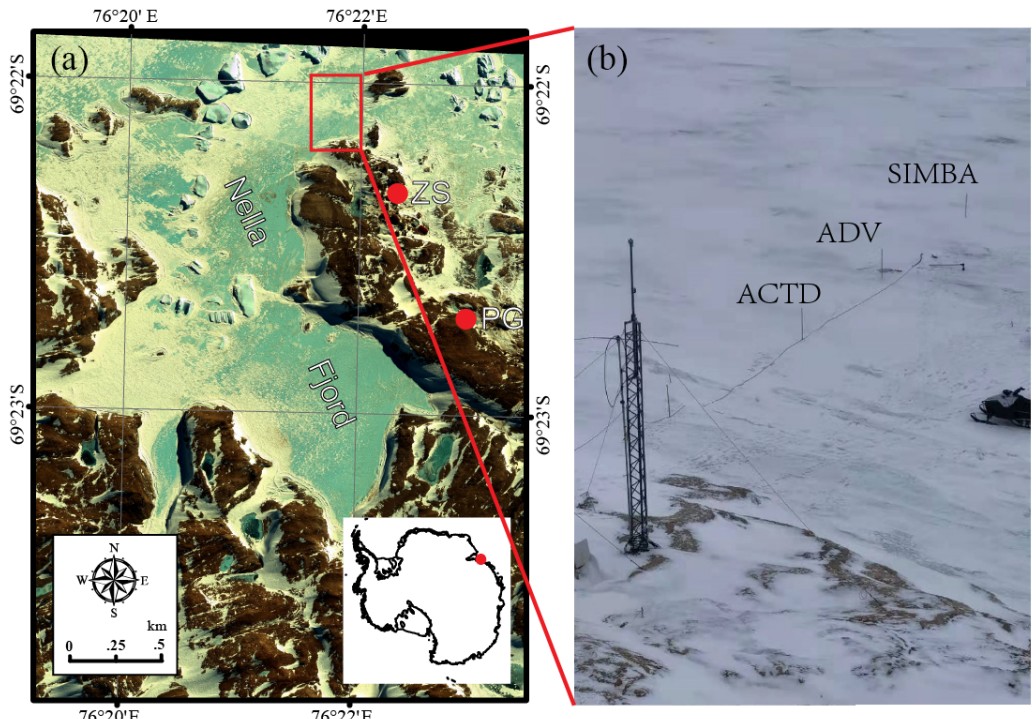


**Figure 1. (a) Satellite image of the observation site in Nella Fjord near Zhongshan Station, modified from the WorldView–2 multi-**
**bands image taken on October 20 2012 (https://worldview.earthdata.nasa.gov); (b) Photo of the observation site shot down from a**
**30-m high slope on April 12, 2021, by Jinkai Ma, one of the co-authors. The photo is not planar as the red box in (a) because of the**
**angle of the shot. The distances among ACTD, ADV and SIMBA were about 5 meters.**
**2.2 Satellite and reanalysis products**
To further investigate the large-scale influences, satellite and reanalysis products were used. The advanced microwave
scanning radiometer 2 (AMSR2) sea ice concentration based on the Arctic Radiation and Turbulence Interaction Study
(ARTIST) sea ice (ASI) algorithm developed at the University of Bremen (https://seaice.uni-bremen.de/sea-ice-
concentration/amsre-amsr2/) was adopted to obtain the percentage of open water in Prydz Bay. These data are updated daily
and have a spatial resolution of 6.25 km (Spreen et al., 2008). The Operational Mercator global ocean reanalysis products,
produced by the Copernicus-Marine Environment Monitoring Service (CMEMS), provide the daily and monthly ocean
currents and mixed layer depth of the global ocean with a 1/12 degree spatial resolution and 3-hour frequency (for more
information, see https://catalogue.marine.copernicus.eu/documents/QUID/CMEMS-GLO-QUID-001-030.pdf, last access:
February 24, 2023). To facilitate comparative analysis, in this study, the nearest neighbour method was employed to interpolate
the CMEMS products to the same projection and spatial resolution as the AMSR2 sea ice concentration.

## 2.3 Oceanic heat flux estimation methods

### 2.3.1 Residual method

The residual method was adapted from the classical Stefan Law. By obtaining measurements of the ice vertical temperature profiles and ice bottom growth or ablation, the residual method has been widely used to estimate the oceanic heat fluxes in previous studies (McPhee and Untersteiner, 1982; Lytle et al., 2000; Perovich and Elder, 2002; Purdie et al., 2006; Lei et al., 2010; Zhao et al., 2019). At the bottom of the sea ice, the heat balance can be expressed by an equilibrium equation as follow:

$$F_w = F_c + F_l + F_s, (1)$$

where $F_w$ is the heat flux from the ocean to the sea ice, $F_c$ is the heat conduction flux through the sea ice, $F_l$ is the latent heat flux caused by the freezing or melting of the ice, and $F_s$ is the specific heat flux generated by the change in the ice temperature. In Eq. (1), the signs of the melting, heating, and upward heat flow are positive, while the signs of the cooling, freezing, and downward heat flow are negative.

The three heat flux terms can be further expressed as follows (Semtner, 1976; Lei et al., 2014):

$$F_c = k_i \frac{T_0 - T_f}{H}, (2)$$

$$F_l = -\rho_i L_i \frac{dH}{dt}, (3)$$

$$F_s = \rho_i c_i \Delta H \frac{dT}{dt}, (4)$$

where $k_i$ is the thermal conductivity of the sea ice; $T_0$ is the temperature of the ice in the reference layer (details are provided in Section 3.4); $H$ is the corresponding sea ice thickness; $T_f$ is the freezing point; $\rho_i$ is the density of the ice; $L_i$ and $c_i$ are the latent and specific heat capacity of the sea ice; $\Delta H$ is the sea ice thickness of the reference layer; $dH/dt$ is the ice growth rate; and $dT/dt$ is the change in the sea ice temperature (Untersteiner, 1961; Millero, 1978; McPhee and Untersteiner, 1982; Lei et al., 2010). The density and salinity of the landfast ice used in this study were 910 kg m$^{-3}$ and 4 psu based on previous observations reported by Lei et al. (2010). $k_i$, $L_i$, and $c_i$ are functions of the salinity and temperature of the ice, and $T_f$ is a function of seawater salinity. These parameters were re-estimated based on the CTD observations. The vertical ice temperature gradient, ice growth/melt rate, and ice temperature changes were calculated from the SIMBA observations.

### 2.3.2 Bulk parameterization method

The oceanic heat flux can be determined from direct measurements of the high-frequency current velocity, temperature, and salinity in the mixed layer in the upper ocean beneath the ice cover in order to evaluate the turbulent heat flux at the ice–ocean interface, which is called the turbulent parameterization method (McPhee, 1992; McPhee et al., 2008). The oceanic heat flux $F_w$ from the ocean mixed layer to the bottom of the sea ice can be expressed as follow (Guo et al., 2015):

$$F_w = \rho_w c_w \langle w'T' \rangle, (5)$$

where $\rho_w$ and $c_w$ are the density and specific heat capacity of the ocean mixed layer; and $\langle w'T' \rangle$ is the turbulent heat flux. The heat transferred from the ocean to the ice depends on both the turbulent stress at the ice–ocean interface (characterized by the frictional velocity $u_0^*$ as the square root of the kinetic stress at the interface) and the effective heat content of the fluid in the turbulent boundary layer, which is roughly proportional to the deviation of the ocean temperature above the freezing point (McPhee, 1992; McPhee et al., 1999; Kirillov et al., 2015). Therefore, the turbulent heat flux can be further parameterized as follow:

$$\langle w'T' \rangle = c_H u_0^* \Delta T, (6)$$

where $c_H$ is the Stanton number of heat exchange efficiency; $\Delta T$ is usually expressed as the difference between the ocean temperature and the freezing point; and $u_0^*$ is the friction velocity at the interface. For the boundary layer beneath the sea ice, the Stanton number $c_H$ is usually assumed to be a constant value of 0.0057 (McPhee, 2002). Therefore, Eq. (5) can be expressed as follow:

$$F_w = \rho_w c_w c_H u_0^* \Delta T. (7)$$

Owing to the roughness beneath sea ice and the fact that the data lack an ocean velocity profile, the friction velocity $u_0^*$ is usually parameterized using the law of quadratic resistance related to the free-stream current. In this study, three different bulk parameterization methods were used to estimate the friction velocity (Table 1). $V$ is the absolute flow velocity relative to the motionless landfast ice, which was observed by the ADV in this study. The velocity perturbation $u'$, $v'$, and $w'$ were estimated by removing the mean from the original time series with 15-minute windows.

Table 1. Three different parameterizations of the friction velocity $u_0^*$

| Parameterizations | Friction velocity equations | References |
|---|---|---|
| Bulk A | $u_0^* = (\langle u'w' \rangle^2 + \langle v'w' \rangle^2)^{1/4}$ (8) | Sirevaag, 2009 |
| Bulk B | $u_0^* = \sqrt{0.0055 * V^2}$ (9) | Kirillov et al., 2015 |
| Bulk C | $u_0^* = \sqrt{0.0104 * V^{1.78}}$ (10) | McPhee, 1979 |

## 3 Results

### 3.1 Snow and ice evolution

Figure 2a shows the SIMBA observations from April 16 to November 7, 2021. The serial numbers of the thermistors start from the deep end of the string in the ocean. Sensor NO. 180 represents the initial location of the ice surface on April 16 when the SIMBA was deployed in the field (dotted lines in Fig. 2). Typically, the sensors above the dotted lines were located in the air and their temperature data exhibited significant daily variations. The sea ice temperature exhibited an obvious gradient of 0.11–0.24°C cm$^{-1}$. The ocean temperature was stable, ranging from −1.7°C to −1.9°C, which was close to the freezing point.

The bottom of the ice (dashed lines in Fig. 2) was identified through visual interpretation according to the method of Zhao et
al. (2017). The ice surface did not experience obvious changes during the cold season, and therefore, the changes in the ice
thickness mainly occurred at the bottom of the ice. The landfast ice was 44 cm thick on the first observation day (April 16),
continued to freeze from May to mid-October, and reached the maximum thickness of 142 cm on October 22. After this, the
bottom of the ice began to melt at a mean rate of $-0.4\pm0.2$ cm d$^{-1}$ until the end of the observation period. The annual mean
growth rate was $0.5\pm0.3$ cm d$^{-1}$, and the maximum daily growth rate was 1.6 cm d$^{-1}$ on May 10, 2021. The monthly mean
growth rate was the largest in May ($0.8\pm0.4$ cm d$^{-1}$) and smallest in October ($0.1\pm0.2$ cm d$^{-1}$), which are similar to previous
observations at Zhongshan Station in 2006 (Lei et al., 2010) and in 2012 (Zhao et al., 2019).

The vertical gradient of the ice temperature profiles shows that snow accumulation on top of the ice cover occurred from May
to August and experienced discontinuous disappearance due to strong winds after September (thin blue lines in Fig. 2b). Finally,
the snow completely disappeared in October when the air temperature rose up to $-2.7°C$. The ice surface began to melt under
the strong solar radiation, and 6–8 cm of sublimation was observed by the SIMBA (thin red lines in Fig. 2b). In particular,
shortly after the SIMBA was deployed, the landfast ice thickness experienced a 4-cm decrease during April 21–26, when the
warm air reached the observation site in the cold winter, and the oceanic heat flux exhibited significant high values during this
period, indicating the influence of the air–ice–ocean interactions on the ice evolution.

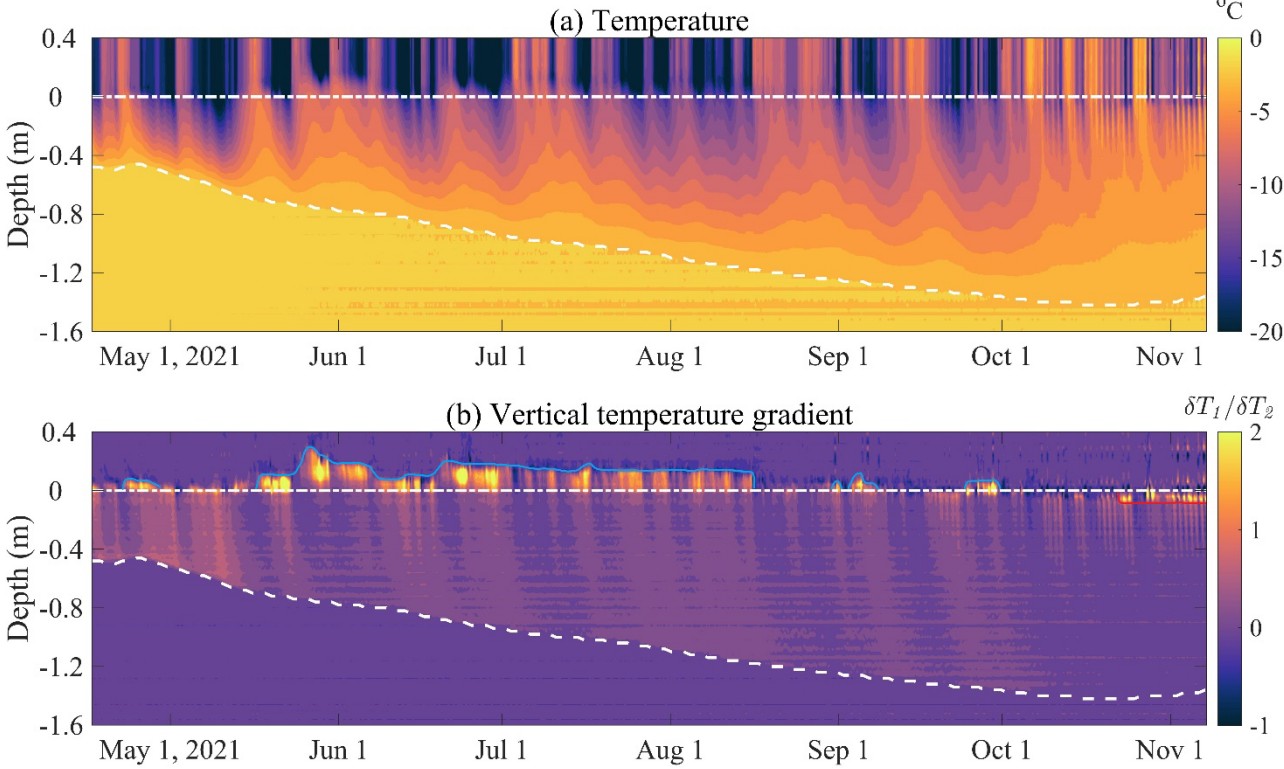


**Figure 2. (a) Temperature profiles and (b) vertical gradient of the temperature profiles recorded by the SIMBA every 6 hours during April–November 2021. The white dashed line and dotted lines in (a) and (b) represent the bottom of the ice and the initial ice surface, respectively. The blue lines and red lines in (b) represent the snow surface and new ice surface after sublimation in summer.**

### 3.2 Ocean temperature, salinity, and density

The times series of the ocean temperature were observed by the CTD deployed 2 m below the surface of the landfast ice. Figure 3a shows the 194 days high-frequency temperature record with a 2-minute interval obtained from April 16 to November 6, 2021. The ocean temperature experienced a rapid increase during April 16–23, from −1.62°C to −1.30°C, and then, it gradually decreased to −1.75°C in the middle of May. In the following months, the ocean temperature remained at around −1.79°C, with a small standard deviation of 0.01°C, until the end of the observations. Therefore, the ocean beneath the ice was relatively warm and was highly variable before the middle of May (−1.64±0.10°C), while the ocean temperature dropped and remained close to the freezing point from then on (−1.79±0.01°C). Based on the spectral analysis, the time series of the ocean temperature exhibited an obvious half-day period, which may be related to the tidal oscillations.

The temperature at the bottom of the sea ice (defined as the mean SMIBA sensor temperature at the lowest 10 cm of the sea ice) was lower than the ocean temperature, indicating that heat was transferred from the warm water to the cold sea ice and

inhibited ice growth at the bottom of the ice. During April–May, the temperature at the bottom of the sea ice exhibited large
variations (−5 to −2.5°C) in response to the variations in the air temperature when the ice was thin and nearly no snow existed.
After the thick snow cover formed, the temperature at the bottom of the sea ice became steady (−2 to −3°C) from June to
November, and the ocean temperature remained stable at around −1.8°C. In particular, the SIMBA recorded a basal ice melting
of 4 cm during April 16–26. This event was accompanied by a concurrent increase in both the air temperature and ocean
temperature, suggesting a heightened transfer of heat from both the air and ocean to the sea ice.

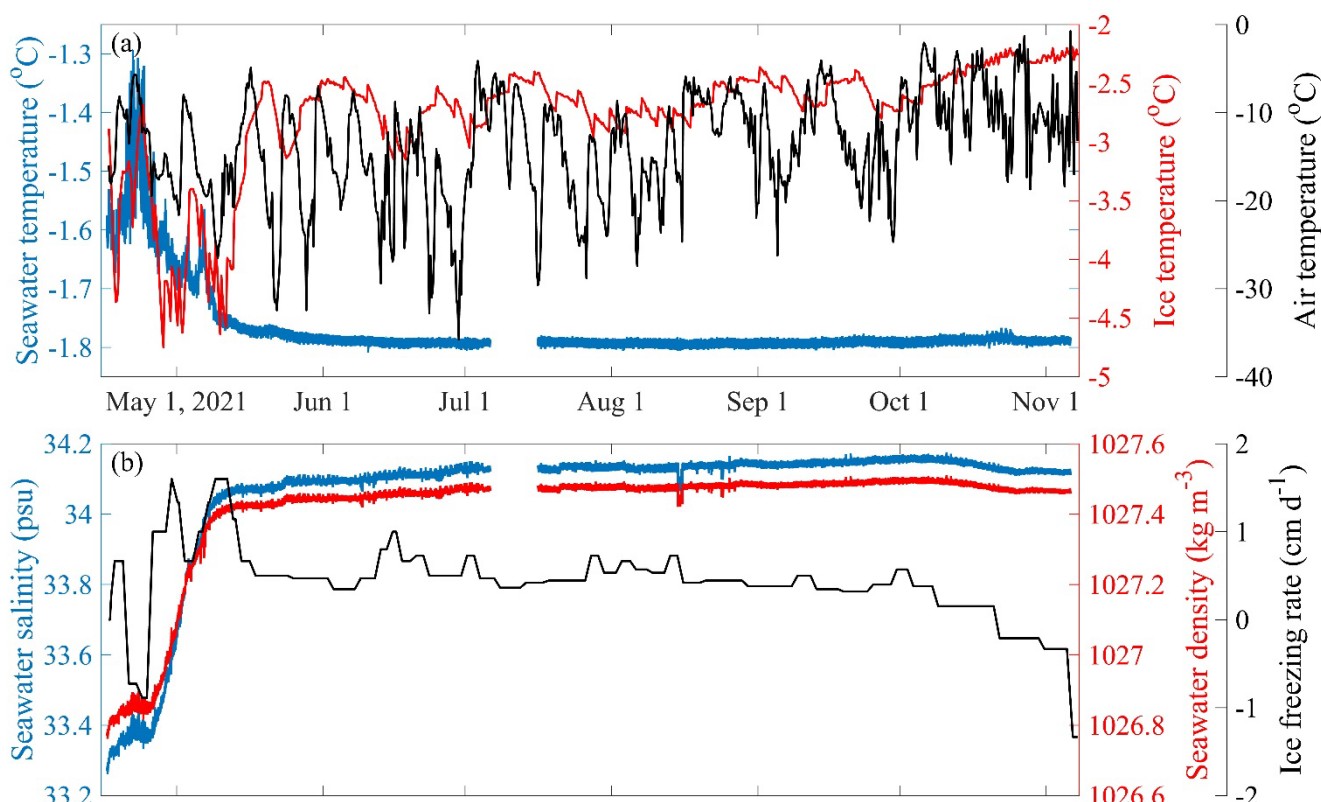

**Figure 3. (a) The seawater temperature observed by the CTD at 2 m beneath the landfast ice surface (blue lines), the ice temperature**
**at the bottom (red lines; defined as the mean temperature derived by the SMIBA sensor located 0.1 m above the bottom of the ice),**
**and air temperature observed by the SIMBA at 1 m above the landfast ice surface. (b) The seawater salinity observed by the CTD**
**(blue lines), the seawater density calculated from the observed temperature and salinity (red lines), and the ice freezing rate at the**
**bottom (black lines) observed by the SIMBA from April 16 to November 7.**
The seawater salinity experienced a rapid increase from 33.34 psu in April to 34.08 psu in May, which was related to the salt
rejection process caused by the high freezing rate of 1.1±0.3 cm d$^{-1}$ at the bottom of the ice (Fig. 3b). More specifically, from
April 19 to 23, the seawater salinity experienced a short period of decrease, different from the long and quick increasing trend,
which may have been related to the slowdown of the freezing at the bottom of the ice during this period due to the obvious
warming of the air and ocean (Fig. 3a). From then on, the seawater salinity (around 34.13±0.02 psu) largely remained stable
with small daily and seasonal deviations. This corresponded with the occurrence of a relatively large and stable freezing rate
at the bottom of the ice (around 0.5±0.2 cm d$^{-1}$) until the middle of October. When the warm season began, the bottom of the
sea ice started to melt at a mean rate of −0.4±0.3 cm d$^{-1}$ (from the middle of October to the middle of November), and the
seawater salinity slightly decreased, indicating that the salt rejection became weaker.

As a function of the seawater temperature and salinity, the seawater density was calculated using the observations measured
by the CTD and the equation proposed by Millero and Poisson (1981). The seawater density exhibited a trend similar to that
of the seawater salinity, which increased significantly during the early winter, with a mean trend of 0.03 kg m$^{-3}$ day$^{-1}$ (Fig.
3b). In the following observation period, the seawater density was stable, with a mean value of 1027.5±0.02 kg m$^{-3}$.
**3.3 Ocean current**
The 3-D current velocity in the meridional (U), zonal (V), and vertical (W) directions at 5 m beneath the surface of the landfast
ice was obtained by the ADV every 40 seconds. A rose diagram of the 2-minute records of the horizontal current is shown in
Fig. 4. The 2-minute frequency records of U and V exhibited large oscillations, mainly varying within ±20 cm s$^{-1}$. In particular,
97% of the U values and 96% of the V values were within ±10 cm s$^{-1}$. W exhibited relatively small oscillations, mainly within
±4 cm s$^{-1}$, and 98% of the W values were within ±2 cm s$^{-1}$. The typical periods of U, V, and W were all half-day periods.

The domain direction was SEE (120°)–SWW (240°), and 79% of the velocity measurements were within 5–15 cm s$^{-1}$ (Fig.
4a). The horizontal velocity was relatively small in April, less than 10 cm s$^{-1}$, and it gradually increased to the maximum value
in June when 75% of the velocity measurements were greater than 10 cm s$^{-1}$. From then on, the horizontal current exhibited a
similar distribution in all three directions, while the range of the dominant velocity changed from 10–15 cm s$^{-1}$ to 5–10 cm s$^{-1}$
(Figs. 4b–i). The horizontal speed exhibited an annual mean velocity of 9.5±3.9 cm s$^{-1}$ and a maximum velocity of 29.8 cm
s$^{-1}$ for the 2-minute interval records.

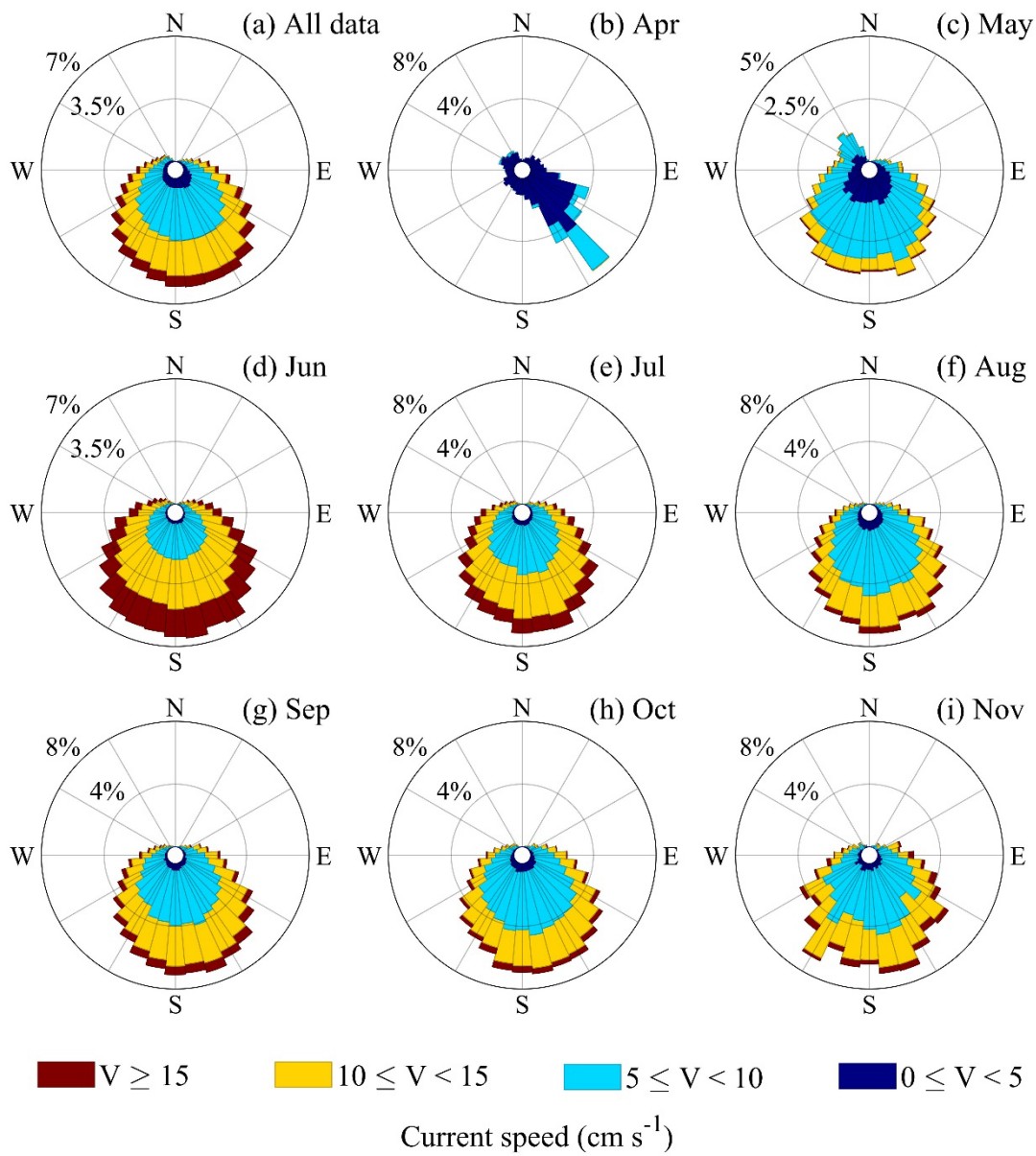

**Figure 4. Roses diagram of the horizontal current speed with a 2-minute resolution for (a) the total time series and (b-i) different months. The different colours represent the different ranges of the current speed. Due to technical issues, only 8 days were available in April and 20 days in May. Please note that the percentage scales are different in the different sub-panels.**

## 3.4 Oceanic heat flux

In the residual method, the vertical gradient of the sea ice temperature is a key term for calculating the conductive heat flux ($F_c$). Under cold and snow-free conditions, the surface air temperature and freezing point are usually used to calculate the

vertical gradient (Lei et al., 2010; Zhao et al., 2019). However, in thick snow or warm cases, the vertical temperature profile
of the sea ice is not linear. In this study, a reference layer close to the bottom of the ice was used to calculate the vertical
gradient to avoid nonlinear biases. McPhee and Untersteiner (1982) set the reference layer at 0.4 m above the bottom of the
ice. Perovich and Elder (2002) set the reference layer at 0.4–0.8 m above the bottom of the ice for different ice thickness
conditions. Lei et al. (2014) set the reference layer at 0.4–0.7 m above the bottom of the ice. In this study, we defined the
reference layer as 0.2 m above the bottom of the ice, and the mean vertical gradient was calculated using the 2 cm interval
temperature profile obtained by the SIMBA.

In previous studies, the empirical value of the freezing point was usually used, but a practical value is more realistic in the $F_c$
calculation. Based on the seawater salinity observations recorded by the CTD, the freezing points were estimated following
the equation derived by Millero (1978). During the observation period, the freezing point was around −1.83°C in April,
gradually decreased to −1.87°C in June, and remained at this value until November.

Figure 5 shows the heat fluxes calculated using the residual method. The variation in the latent heat flux ($F_l$) was strongly
correlated with the growth and ablation of the sea ice. During the study period, $F_l$ was positive in the cold season, except for a
short melting period in April. During April 21–24, due to the influences of the warm air and ocean, the SIMBA recorded
obvious melting at the bottom of the ice and $F_l$ exhibited a negative value of −20 W m$^{-2}$. In October, the melt season began
and $F_l$ became negative. The specific heat flux $F_s$ was smaller throughout the study period, oscillating around 0 W m$^{-2}$. The
conductive heat flux $F_c$ was relatively large before the middle of May (up to −80 W m$^{-2}$), gradually decreased to −20 W m$^{-2}$
in September, and finally reached −10 W m$^{-2}$ in October and November. The oceanic heat flux exhibited a larger value of
41.3±9.8 W m$^{-2}$ in April and then decreased to around 10 W m$^{-2}$ from June to October, but it quickly increased to 50 W m$^{-2}$
in November before the observation period ended. The annual mean oceanic heat flux for the entire study period was 12.2±10.9
W m$^{-2}$.

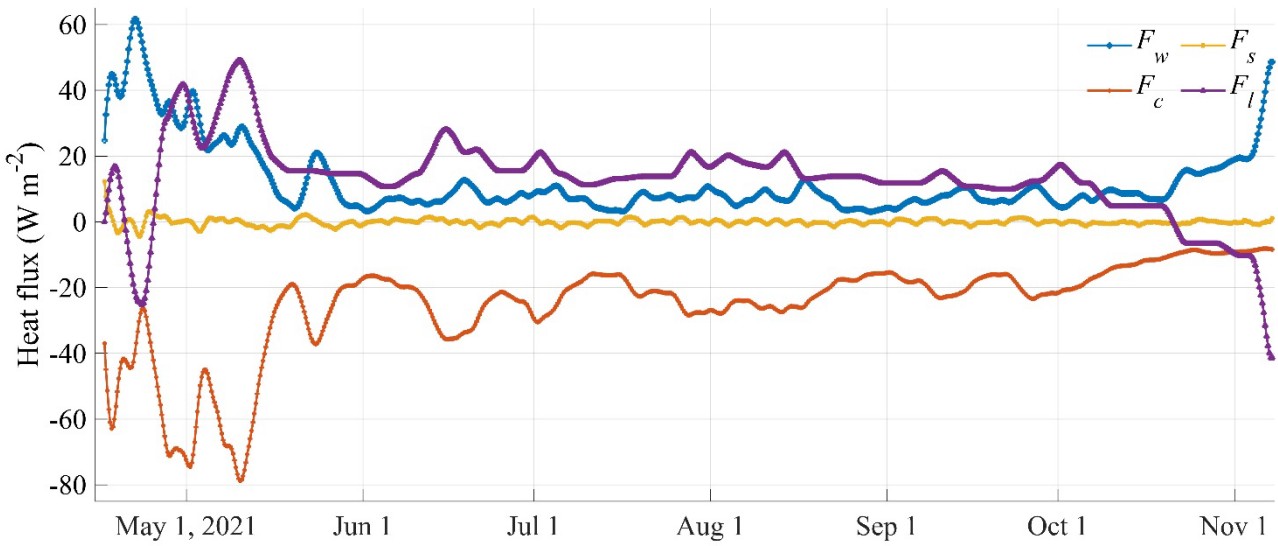


**Figure 5. Conductive heat flux ($F_c$), latent heat flux ($F_l$), specific heat flux ($F_s$), and oceanic heat flux ($F_w$) were estimated using the residual method and a reference layer located 0.2 m above the bottom of the ice. The time interval is 6 hours.**

In contrast to the residual method, previous studies have developed bulk parameterization methods for calculating the oceanic heat flux when the observations of ocean parameters are available (McPhee, 1979, 1992; Sirevaag, 2009; Kirillov et al., 2015). In this study, the ocean velocity, temperature, and salinity in the ice–ocean boundary layer were recorded at a high frequency by the ADV and CTD, which provided a chance to evaluate the oceanic heat flux using bulk parameterization methods.

During the observation period, the ocean temperature was always warmer than the freezing point, indicating that the heat flux was from the ocean to the ice. The temperature difference ($\Delta T$) between the ocean and the freezing point was 0.26±0.08°C in April and decreased gradually to 0.08°C from June to November. Three different bulk parameterization methods were used in this study (Bulk A: Sirevaag, 2009; Bulk B: Kirillov et al., 2015; Bulk C: McPhee, 1979), and their main differences were due to the expressions of the fractional velocity and empirical parameters (Table 1).

The hourly oceanic heat flux values calculated using three bulk parameterization methods exhibit variations similar to that of the results of the residual method, that is, high values of 60–80 W m$^{-2}$ in April and then gradually decreasing to 10–30 W m$^{-2}$. The annual mean oceanic heat flux values were 19.7±5.3 W m$^{-2}$, 13.6±3.1 W m$^{-2}$, and 24.4±5.4 W m$^{-2}$ for the Bulk A, Bulk B, and Bulk C methods, respectively, and 12.2±10.9 W m$^{-2}$ for residual method (Fig. 6a). The values obtained using the bulk methods were 9.0±8.9 W m$^{-2}$ larger on average than that obtained using the residual method during the study period.

According to the monthly oceanic heat flux trends shown in Fig. 6b, the oceanic heat flux values were 18.4 W m$^{-2}$, 15.7 W m$^{-2}$, 31.4 W m$^{-2}$, and 41.3 W m$^{-2}$ in April for the Bulk A, Bulk B, Bulk C, and residual methods, respectively. In addition, the

oceanic heat flux had large standard deviations in April, 10–20 W m$^{-2}$ for the bulk methods and 10 W m$^{-2}$ for the residual
method, indicating a large variation in the hourly time series. From May to October, the standard deviations were generally
less than 5 W m$^{-2}$. Among the three bulk parameterization methods, the results of the Bulk C method were relatively larger
than those of the Bulk A and B methods.

Previous studies estimated the oceanic heat flux under landfast ice in Prydz Bay using different methods. Allison (1981)
estimated the oceanic heat flux near Mawson Station from monthly mean temperature and ice growth data. In the early stage
of sea ice growth, the thermohaline convection caused by the brine rejection made the flux very high, and it could reach 50 W
m$^{-2}$. Heil et al. (1996) used a multilayer thermodynamic model to simulate sea ice growth at Mawson Station. The multi-year
average oceanic heat flux estimated from daily values was 7.9 W m$^{-2}$, and the annual mean was 5–12 W m$^{-2}$ from 1958 to
1986. Lei et al. (2010) estimated the oceanic heat flux near Zhongshan Station in early April to be 15–20 W m$^{-2}$. Yang et al.
(2016) estimated the oceanic heat flux to be 25 W m$^{-2}$ in March–April using a thermodynamic model. According to weekly
observations near Zhongshan Station, Zhao et al. (2019) interpolated and calculated the daily oceanic heat flux from March to
May to be 30 W m$^{-2}$. In this study, the average oceanic heat flux calculated using the residual method and the bulk methods
are consistent with those of previous studies on the seasonal scale, and the quantitative difference may be related to the specific
methods and environmental parameters for the given years. In this study, we utilized a higher temporal resolution (6 hours for
the residual method and 2 minutes for the bulk methods), which provide more details and insights for the readers and
communities, while the estimation of the oceanic heat flux at the bottom of the ice based on the residual method may produce
great errors within a short time window (Lei et al., 2010).

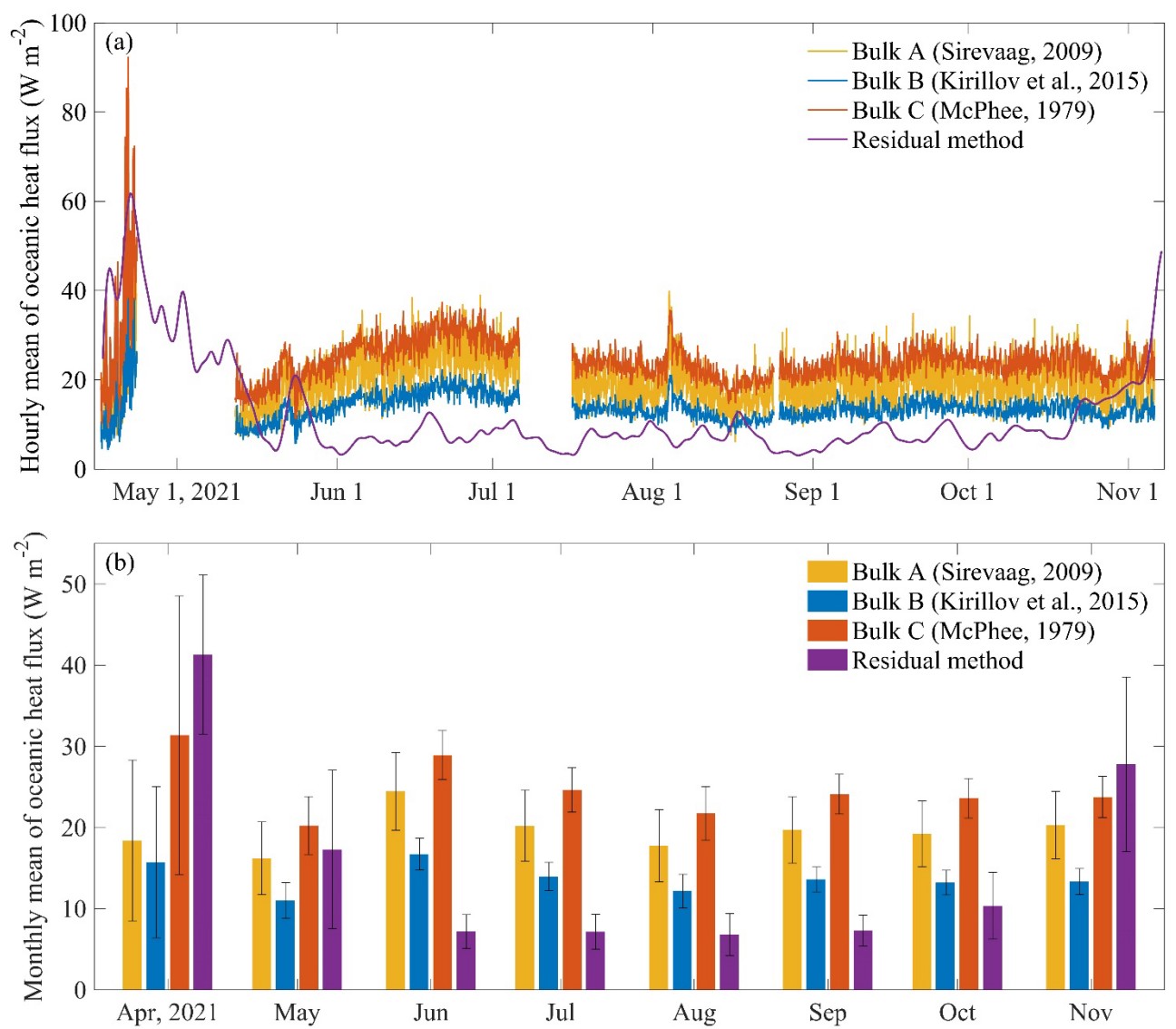


Figure 6. (a) Hourly mean $F_w$ was calculated using the three bulk parameterization methods and the 6-hourly mean $F_w$ was calculated
using the residual method and (b) the monthly mean $F_w$. The error bars in (b) represent ±1 standard deviation of the hourly mean
values.

Table 2. Inter-comparisons of the mean ± standard deviation of the oceanic heat flux values (W m$^{-2}$) were calculated using different
methods.

| Methods | Apr. | May | June | July | Aug. | Sept. | Oct. | Nov. | Total |
|---|---|---|---|---|---|---|---|---|---|
| Residual method | 41.3±9.8 | 17.3±9.8 | 7.2±2.1 | 7.2±2.2 | 6.8±2.6 | 7.3±1.9 | 10.4±4.1 | 27.8±10.8 | 12.2±10.9 |
| Bulk A | 18.4±9.9 | 16.2±4.5 | 24.5±4.8 | 20.2±4.4 | 17.8±4.4 | 19.7±4.1 | 19.2±4.0 | 20.3±4.2 | 19.7±5.3 |

| | | | | | | | | |
|---|---|---|---|---|---|---|---|---|
| Bulk B | 15.7±9.3 | 11.0±2.2 | 16.7±2.0 | 14.0±1.7 | 12.2±2.1 | 13.6±1.5 | 13.2±1.5 | 13.2±1.6 | 13.6±3.1 |
| Bulk C | 31.4±17.2 | 20.2±3.6 | 28.9±3.0 | 24.6±2.7 | 21.8±3.3 | 24.1±2.5 | 23.6±2.4 | 23.8±2.5 | 24.4±5.4 |

## 4. Discussion

The minute-frequency annual observations of variables in the ice-ocean interface in this study provide a clear picture of how they varied on an hourly, daily or seasonal scale, and fill up the data gap in Zhongshan Station. As the relative studies in other regions, those variables may be affected by the short-term cycle of sub-glacial current (McPhee et al., 1996) and ocean tide current (Lei et al., 2010). To further enrich our analysis, the relationships between processes on the local scale and pan-Prydz Bay scale were discussed here.

### 4.1 Potential influences of local tidal oscillations

The local tides may influence the evolution of sea ice (Lei et al., 2009). The tidal oscillations were reconstructed using the harmonic analysis method (Pan et al., 2018) and the harmonic constants from E et al. (2013). In this study, the periodogram method (Welch, 1967) was used to detect the periodicity of the long time-series observation data. Power spectrum analysis of the signal revealed that the tidal oscillations exhibited two peaks. The largest peak had a period of 1 day, and the second largest peak had a half-day period, indicating that the tide near Zhongshan Station was an irregular diurnal tide (Fig. 7). To further investigate the relationships between the tidal oscillations and oceanic variables, the same spectral analysis was employed for all of the observed ocean variables. The ocean temperature exhibited the largest peak with a period of 1 day and a relatively low peak with a half-day period. In contrast, the seawater salinity, U, V, W, and the results of the three bulk parameterization methods exhibited the largest peak with a half-day period and a relatively low peak with a 1-day period (Fig. 8). The results of the spectral analysis indicate that the ocean temperature, salinity, U, V, W, and oceanic heat flux were greatly affected by the tidal oscillations.


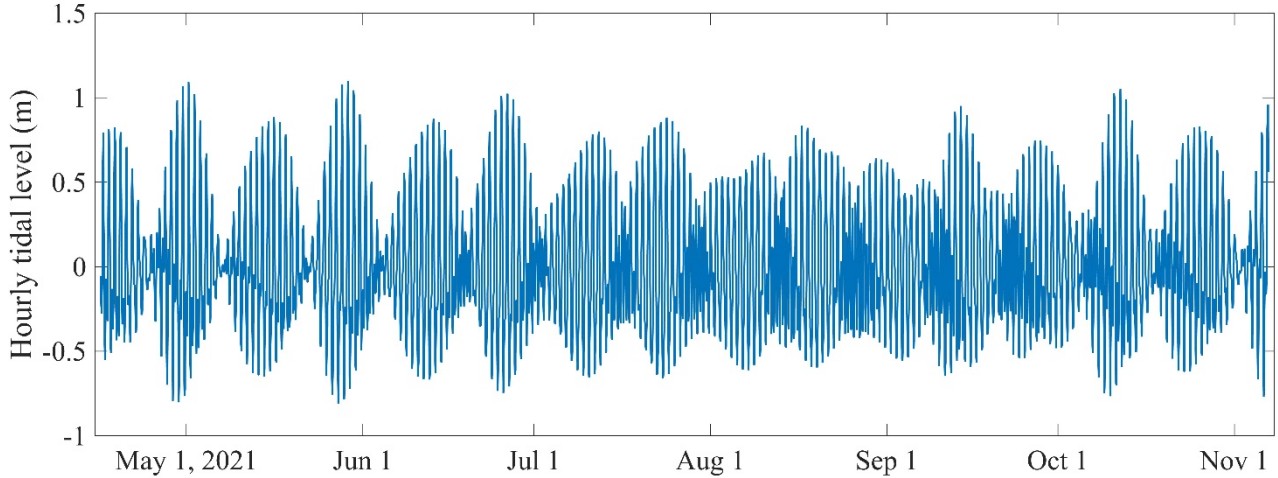


**Figure 7. The tidal oscillations were constructed using the harmonic analysis method (Pan et al., 2018) and the harmonic constants**
**of E et al., (2013). The temporal resolution of this dataset is 1 hour.**


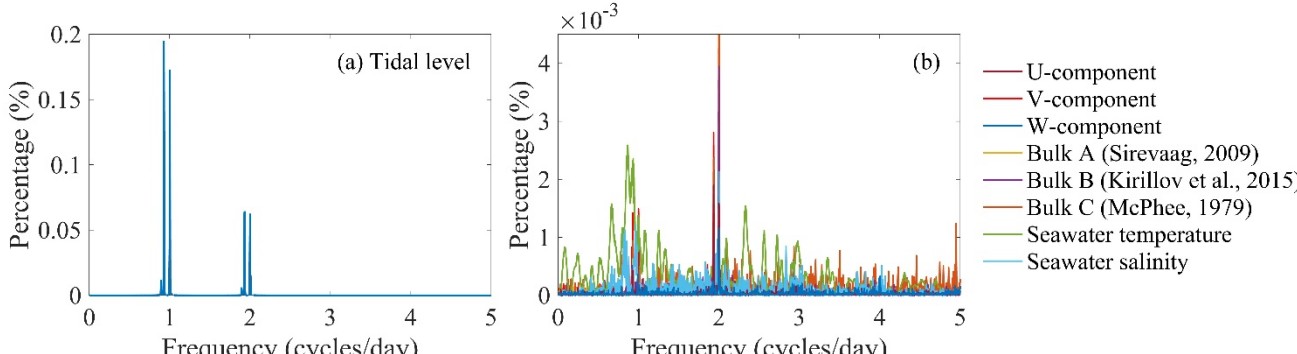


**Figure 8. (a) The results of the spectral analysis of the tidal oscillations and the observed ocean variables, and (b) the calculated $F_w$.**
**The periodogram method was used to detect the periodicity (Welch, 1967).**
In April, the observed seawater temperature and salinity exhibited a special pattern, that is, the water was relatively warm and
fresh in the equilibrium tide state, while it was cold and salty in the low and high tide states (Figs. 9a, b), which may have been
related to the efficient horizontal heat transport when the surrounding area was not completely covered by ice. However, in
the other months, the larger observed vertical velocity enhanced the vertical mixing, and therefore, no significant variations in
the seawater temperature and salinity and the oceanic heat flux were observed during the same period.

Furthermore, when the tide level changed from low to high, the hourly U changed from a slightly positive distribution (0.7±1.2
cm s$^{-1}$) to a deeply positive distribution (1.2±1.1 cm s$^{-1}$), indicating predominantly eastward flow during the high tide level
conditions (Fig. 9c). V changed from a slightly negative distribution (−1.3±1.6 cm s$^{-1}$) to an intensely negative distribution
(−2.1±1.3 cm s$^{-1}$), suggesting that the southward flow became stronger when the tide level was high (Fig. 9d). W did not vary
prominently, and the mean values were almost the same, 0.2±0.3 cm s$^{-1}$ and 0.2±0.2 cm s$^{-1}$during the low and high tide levels,
respectively (Fig. 9e).

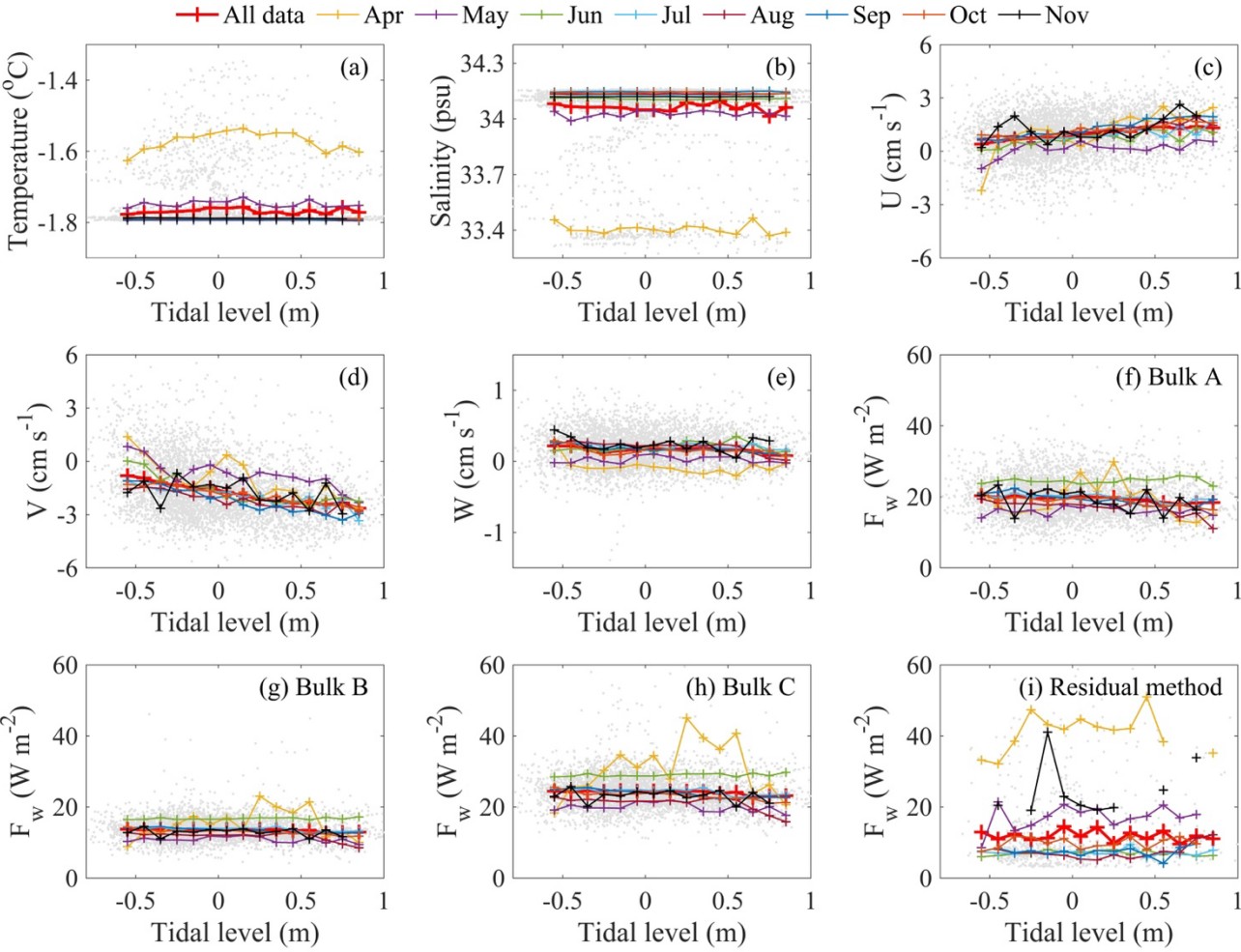


**Figure 9. Scatter plots of the tidal level versus the oceanic variables: (a) seawater temperature, (b) seawater salinity, (c) U-component**
**velocity, (d) V-component velocity, (e) W-component velocity, and (f–i) $F_w$ from the Bulk A, B, C, and residual methods. The grey**
**dots are the hourly mean values of the variables, and the different lines represent the monthly mean values for 0.1 m tidal level bins.**
**4.2 Relationships between large-scale and local phenomena**
Prydz Bay was covered by sea ice in the cold season. Ice floes appeared widely in March, and landfast ice started to form one
month later in April near Zhongshan Station. From May to October, ice floes completely covered Prydz Bay, except for several
large polynyas (Fig. 10d), for example, Davis Polynya (DaP) and the Four Ladies Bank Polynya (FLBP) on the east side and
the Mackenzie Bay Polynya (MBP) and Cape Darnley Polynya (CDP) on the west side (Hou and Shi, 2021; Nihashi and
Ohshima, 2015; Williams et al., 2016). In addition, the landfast ice gradually extended to around 100 km along the zonal
direction. In November, the ice floe concentration decreased, and the landfast ice cover reached the maximum extent (Fig. 10).
The open water area accounted for nearly 80% of the entire ocean grid in March, allowing more solar heat flux to be absorbed
by the ocean, which was the energy basis for the warm ocean in April (Fig. 11). The large-scale circulation in Prydz Bay
indicated the existence of a westward current along the Antarctic coastline, which was stronger in the ice-free and low ice
concentration months and weaker in the high ice concentration months. In April, the large-scale current carried the warm water
from low latitudes to high latitudes, contributing to the observed rise in the ocean temperature near Zhongshan Station. From
then on, the large-scale current weakened, and the horizontal heat transport decreased.

The four large polynyas shown in Fig. 10d started to form in April, which led to the release of a large amount of salt through
new ice production during their existence. As a result, the ocean mixed layer in the corresponding locations derived from
Mercator global ocean reanalysis products exhibited obvious thickening from May to October (figure not shown). In addition,
the thickening of the entire ice region in Prydz Bay contributed to the strengthened vertical mixing caused by the salt rejection
as the sea ice continued to grow. The high seawater salinity observed by the CTD near Zhongshan Station (yellow lines in Fig.
11) confirms this assumption. Considering the reduced horizontal heat transport, the evolution of the ocean temperature was
mainly affected by local factors. In this study, the observations were conducted close to the shore at a water depth of around
10 m, making full mixing of the shallow water possible. Therefore, the seawater temperature remained at a stable level from
June to November (red lines in Fig. 11).

The water depth near the shoreline may have affected the vertical mixing capacity. The observations of the seawater
temperature from the SIMBA sensors at 2 m beneath the ice surface and the CTD were obviously different (annual mean
difference of $-0.17\pm0.03$°C), which was largely beyond the errors of the instruments. The water depths of the SIMBA and
CTD sensors were 4.5 m and 13 m, respectively, and this difference is believed to have caused the different vertical mixing
strengths and thus the different seawater temperatures.

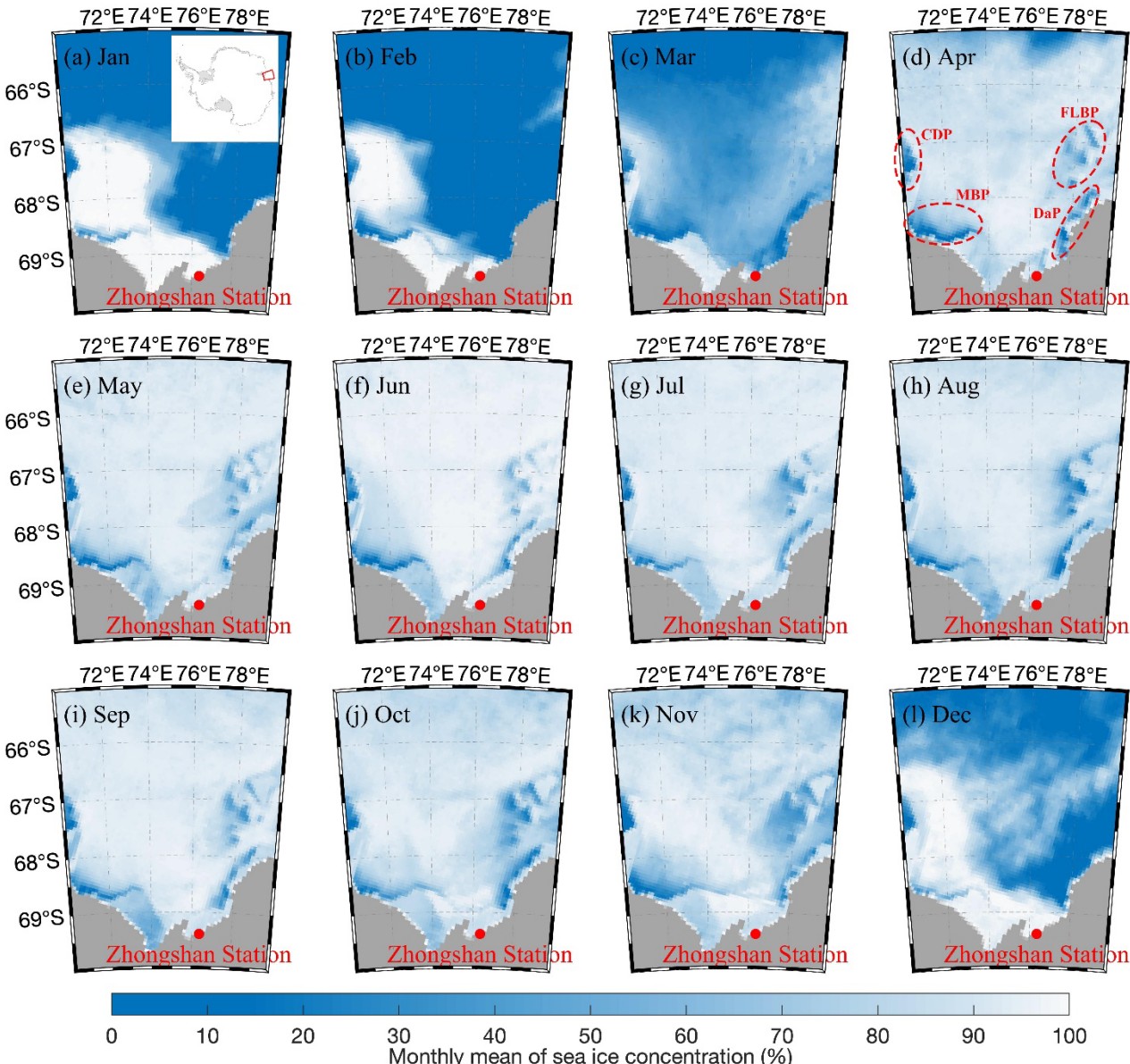


**Figure 10. (a–i) Evolution of the monthly sea ice concentration in Prydz Bay from January to December 2021. The domain of Prydz Bay (70–80°E, 65–70°S) in Antarctica is shown in the right-top corner of (a). The sea ice concentration dataset was retrieved from the AMSR2 product provided by Bremen University (https://seaice.uni-bremen.de), with a spatial resolution of 6.25 km. The locations of four large Polynyas are marked in (d), i.e., the Davis Polynya (DaP) and Four Ladies Bank Polynya (FLBP) on the east side and the Mackenzie Bay Polynya (MBP) and Cape Darnley Polynya (CDP) on the west side.**


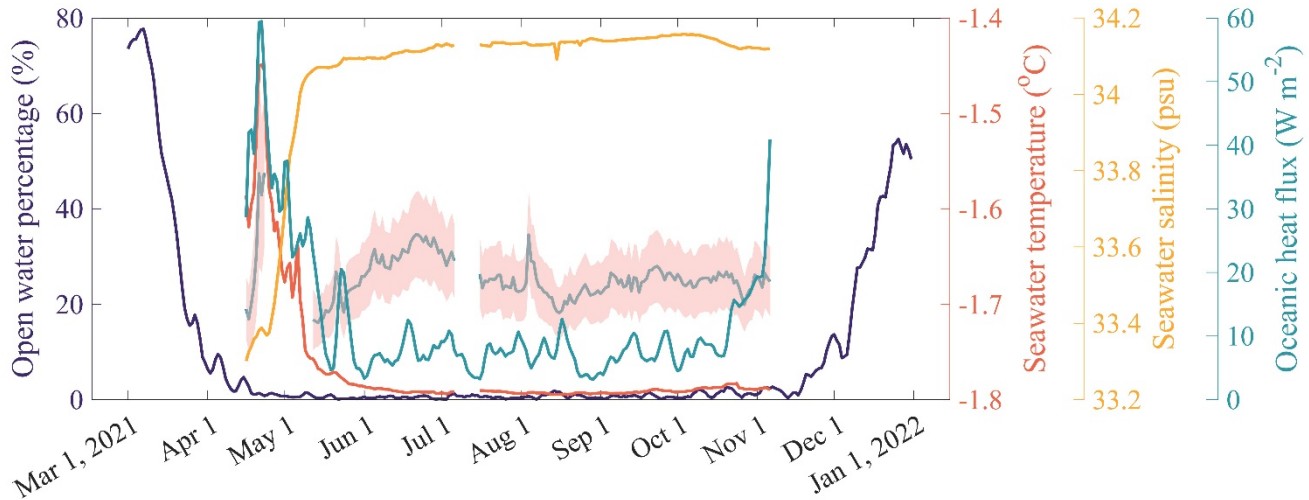


**Figure 11. The time series of the daily percentage of open water (purple lines) relative to the domain of Prydz Bay (shown in Fig. 10)**
**and the seawater temperature (red lines), seawater salinity (yellow lines), mean oceanic heat flux from the Bulk A, B, and C methods**
**(grey lines with rose shading), and the oceanic heat flux from the residual method (green lines). The open water area was defined as**
**the sum of the grid cells where the sea ice concentration was less than 15%. The rose shading indicates ±1 standard deviation.**
**5 Conclusions**
The heat and momentum balances among the air–ice–ocean are some of the most important processes in the polar regions. The
air–ice interactions have been well investigated due to the fact that on-ice observations are relatively easy to conduct. However,
the ice–ocean interactions have rarely been studied due to the difficulty and limitations of underwater observations. The
oceanic boundary layer beneath sea ice plays an important role in the growth and melting of sea ice. In this study, an integrated
ice–ocean observation system, including an ADV, CTD, and SIMBA, was deployed on the landfast ice 1 km far from
Zhongshan Station in Prydz Bay, East Antarctica. The ocean temperature, salinity, and velocity were observed with a 40-
second resolution and 8-month observation period and were investigated for the first time in this region.

The SIMBA temperature chain recorded the vertical temperature profiles of the air–snow–ice–ocean, which were used to
estimate the snow and ice thicknesses and oceanic heat flux using the residual method. The results show that 98 cm of landfast
ice formed from April to October, with a mean growth rate of 0.5±0.3 cm d$^{-1}$; and 4 cm melted in November, with a rate of
−0.4±0.2 cm d$^{-1}$ until the observation period ended. Approximately 6–8 cm of surface sublimation was observed in summer.
The maximum snow thickness was around 30 cm in May and remained at 10–20 cm until August. The CTD recorded the 40-
second resolution seawater temperature and salinity at a depth of 5 m beneath the ice surface. The seawater temperature rapidly
increased from −1.62°C to −1.30°C in April and then gradually decreased to −1.75°C in May. The seawater temperature
remained stable from June to November, with a mean of −1.79±0.01°C. In April, the landfast ice was 44–50 cm thick and the
ice surface was snow free; therefore, the variations in the air temperature exerted a larger influence on the ice and seawater

temperatures. The significant increases in the air and seawater temperatures led to an increase in the temperature of the bottom of the ice, which contributed to the sudden melting of 4 cm from the bottom of the ice observed by the SIMBA. The thick snow cover from May to August provided an isolation layer for the ice and ocean, which contributed to the stability of the seawater temperature during this period.

The seawater salinity increased from 33.34 psu in April to 34.08 psu in May, with a rate of 0.04 psu d$^{-1}$. From June to November, the seawater salinity was stable at around 34.13±0.02 psu. The seawater density calculated from the observed seawater salinity increased from 1026.8 kg m$^{-3}$ to 1027.4 kg m$^{-3}$ from April to May and remained at 1027.5±0.02 kg m$^{-3}$ from then on. The current velocity was recorded by the ADV from April to November. The analysis of the 2-minute resolution time series revealed that 79% of the ocean velocity values were within 5–15 cm s$^{-1}$, and the annual mean was 9.5±3.9 cm s$^{-1}$. The maximum velocity of 29.8 cm s$^{-1}$ was observed on June 25, 2021. The dominant current direction was SEE(120°)–SWW(240°). The spectral analysis results suggest typical half-day periods for U, V, and W, which may be related to the tidal oscillations near Zhongshan Station. The meridional velocity V was dominated by the southward flow and became stronger when the tide level was higher.

The oceanic heat flux was estimated using the residual method and three different bulk parameterization methods. The results exhibit a similar peak of 60–80 W m$^{-2}$ in April–May and a decreasing trend to a stable level of 10–30 W m$^{-2}$ from then on. The annual mean values were 12.2±10.9 W m$^{-2}$, 19.7±5.3 W m$^{-2}$, 13.6±3.1 W m$^{-2}$, and 24.4±5.4 W m$^{-2}$, respectively, for the residual and Bulk, A, B, and C methods. The large differences were mainly caused by the different formulas for the friction velocity, indicating the uncertainties of the empirical equations. The estimated results obtained in this study are consistent with those of previous studies, which were usually based on low-frequency ice thickness observations. The oceanic heat fluxes exhibited similar half-day periods, which are also believed to be related to the tidal oscillations.

The observations of seawater temperature, salinity, U, V, and W and the estimation of the seawater density and oceanic heat flux exhibited periods similar to that of the local tidal oscillations, suggesting that the tides were one of the main drivers of the oceanic variations near Zhongshan Station. The large-scale sea ice distribution and current transformation affected the absorption of solar radiation by the upper ocean and the horizontal heat transport, which was another main driver of the oceanic variations near Zhongshan Station. Both the local and large-scale influences played important roles in the oceanic heat flux and thus the ice–ocean interactions.

In this study, the attainment of high-frequency oceanic measurements provided an opportunity to investigate the details of the ice–ocean interactions beneath landfast ice on the diurnal and seasonal scales. The bulk parameterization was used to estimate the oceanic heat flux near Zhongshan Station, providing more interesting information than the residual method does. The use

of more ice and ocean equipment, such as ice thickness radar, ocean temperature chains, and ice salinity gauges, will be
considered in the future, to fill the remaining data gap.

**Data availability**

The observation data are available from the Science Data Bank. The seawater temperature and salinity recorded from a cable-
type CTD are publicly available at https://doi.org/10.57760/sciencedb.07693 (Zhao and Hu, 2023). The air-ice-ocean
temperature profile derived from Sea Ice Mass Balance Array (SIMBA) is publicly available at
https://doi.org/10.57760/sciencedb.07684 (Zhao and Hu, 2023). The 3-D current velocity 5-m beneath landfast ice recorded
from an Acoustic Doppler Velocimeter (ADV) are publicly available at https://doi.org/10.57760/sciencedb.07692 (Zhao and
Hu, 2023).

**Author contributions**

JC conceptualized this study and designed the numerical methods. HH carried out the experiments and wrote the manuscript.
JC, PH, ZL and FH helped analyse the results and revised the manuscript. JM provided and helped process the sea ice
observation data. XC assisted during the writing process and critically discussed the contents.

**Competing interests**

One of the co-authors is a member of the editorial board of *The Cryosphere*, and the authors have no other competing interests
to declare.

**Acknowledgement**

This study was financially supported by the National Natural Science Foundation of China (42276251, 42211530033,
41876212). PH was supported by the Australian Government through Australian Antarctic Science Projects (no. 4506) and the
International Space Science Institute (grant no. 406).

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
