# Peer review of "Annual evolution of the ice—ocean interaction beneath landfast ice in Prydz Bay, East Antarctica"

_The Cryosphere, 2022_

## Referee Comment (RC1)

**Review on "The diurnal evolution of oceanic boundary layer beneath early-frozen landfast ice in Prydz Bay, East Antarctica" by Hu et al. (2022), submitted to TC Discussions**

*September 15, 2022*

The submitted manuscript describes a data set of ocean temperature, salinity and currents as well as thereof derived ice & ocean properties such as density and heat fluxes. The data record from the presented "integrated ocean observation system" (which contains an Acoustic Doppler Velocimeter, a CTD and a sea ice mass balance array) in Prydz Bay, Antarctica, spans about seven days in total and features measurements with a temporal resolution of 30 seconds to six hours.

Overall, I am missing clear focus points in the manuscript. Besides the comparatively high frequency of ocean temperature and salinity measurements and a fairly comprehensive description of the measurements and derived quantities, I find it hard to spot truly unique findings in the manuscript that would qualify for a swift publication in The Cryosphere. If the authors are able to better relate their measurements to other data sources / the larger scale context, together with a more profound discussion and evaluation of uncertainties & shortcomings, there might still be a chance to reconsider it after several major revisions. Please find my main remarks below under "General comments".

Language & grammar-wise (judging from a non-native English perspective), several parts of the paper (see "Specific comments" below) would most certainly profit from another round of thorough proofreading and rephrasing, especially in relation to overly long sentences and word order.

**General comments**

1) The presented time series of just seven days is particularly short, compared to other similar data sets in both hemispheres. How do you justify the significance of this data record? It is not 100% clear to me how this data set sets itself apart from any other. Further, you only mention an instrument malfunction at the very end of Ch.5, which is likely the cause for the short data record, right? Why is this not mentioned right at the beginning of your data description?!

2) The manuscript often mentions the apparent benefits of "minute resolution" measurements, without clearly differentiating between the different data sources. For instance, the SIMB measurements of vertical temperatures are only available four times a day (hence, imprinting on presented heat fluxes). Please revise the respective parts carefully.

3) The authors apparently decided to leave out a "traditional" chapter on the applied methodology to process and analyze the recorded data. Later in the text in the context of results (Ch.3.6), at least the heat flux calculations are explained. However, I consider these parts as misplaced. I would recommend a new separate chapter on methodical aspects & data processing prior to the results section. In this context, Chapters 3.6.1 and 3.6.2 could also be thoughtfully merged and at the same time streamlined to the most relevant aspects.

4) Almost all figures require a careful overhaul, be it due to the lack of proper labelling, low resolution images, non-barrier-free colormaps or "just" an insufficient / too short caption. You will find more detailed comments on all these below, right after specific comments to individual chapters.

5) There is no further information on the larger scale environmental conditions (sea ice cover, atmospheric / ocean reanalysis, etc.) at all. Even if you omit to directly relate these conditions to your own data, it would be extremely helpful to have those for a proper context.

6) Please pay attention on using a (relevant) number of digits after the decimal point. Often, there is unnecessary detail given, especially when the numbers end with ".0". In addition, try to be consistent throughout the manuscript.

7) You describe your results/measurements in past tense (e.g., "Figure 6 showed") → please use the present tense in that regard.

**Specific comments** *(incl. technical notes)*

*Abstract*

P.1, L.14: "COMPACT-CTD" → there is only an ACTD mentioned in the manuscript. Please explain, also why capital letters are used here.

P.1, L.16: Not all measurements are minute-resolution, right? SIMB → six-hourly

P.1, L.20: "the bulk ..." → "a bulk"

*Ch.1: Introduction*

P.2, L.37: Fraser et al. (2021) would be another good reference here (DOI: 190.51.94/tc-15-5061-2021)

P.2, L.60: "the flux balance equation" → "a flux balance equation"

P.2, L.60: 8 psu salinity → salinity of the sea ice? More specific please.

P.2, L.62: Explain abbreviation "HIGHTSI" and give the respective reference

P.2, L.64: Again – "a residual" instead of "the residual"

P.2, L.65: Leave out ".0"

P.3, L.67: "there are few studies" →so there are some apparently?

P.3, L.75: please give a reference for the "modified Stefan's law", or explain briefly

P.3, L.77: "Based on *measurements* of..."

P.3, L.77: Please explain abbreviations in the text; as they are used for the first time here

P.3, L.78-80: Can be left out → phrasing in its current form

*Ch.2: Observations*

P.3, L.82: Coordinates misplaced; move to beginning of next sentence

P.3, L.84: Reference for the landfast ice cover duration?

P.3, L.86-88: Are these own observations or is the reference missing?

P.3, L.91-96: Please indicate reference papers/reports for the respective measurement devices (could also be moved to a table in general; together with other instrument characteristics)

P.4, L.101: "every five days" → in a seven-day data record, it is sufficient to call that "twice"

*Ch.3: Results*

P.4, L.112-114: Please revise this sentence to improve grammar

P.4, L.114: Be careful with the use of the word "significant", as this usually refers to a statistical significance. If there are no statistics involved here, please try to use a different wording.

P.5, L.120: You write that the ice-water interface was determined by a simple threshold (freezing point temperature of sea water). Can you elaborate more on that topic, especially how you handled noisy data and the given uncertainty for the IMB temperature measurements?

P.5, L.123: "observed in the field" → do you mean direct measurements, for instance by a drill?

P.5, L.125: "in consequence" → a consequence

P.5, L.131: "the ACTD"

P.5, L.131: "2m below the ice surface" → how thick was the ice at this position?! I would assume that at least platelet ice could fairly quickly become a problem for CTD measurements at this depth…

P5, L.134: with "about 0.1m above the ice bottom" – do you mean the lowest 10cm? Please rephrase

P.6, L.138/139: Check grammar and sentence breaks

P.6, L.139/140: Please be more precise here. Which temperature gets warmer, and compared to what/when? Plus: "more heat", not "more heat flux"

P.7, L.147/148: Did you mix up smallest & largest deviation?

P.7, L.148: Add "mean" or "average" before "ocean salinity" and replace "remained at"

P.8, L.157: Remove "internationally recognized"

P.8, L.158-166: Multiple remarks; Please elaborate in more detail why you used this particular equation, i.e., why you consider it as suited for your observations in Prydz Bay. Further, please do not just copy the denoted symbols from the source publication without explaining them first together with the respective units (t, S, rho). Also, be more precise and consistent with the indexing, for instance in case of salinity (ice or water salinity?).

P.8, L.168 & 170: Be careful when referring to a "trend", as this is again mainly used in the context of statistical analysis.

P.8, L.171: Check grammar & sentence breaks

P.9, L.180: "u-component" and "v-component"; check grammar after "It can be seen"

P.9, L.182: "ROSE analysis" is not the correct wording. It's "just" a diagram.

P.10, L.186: Please explain what you mean by "compound current"

P.10, L.187: Please give a proper reference / data citation for the data set from the Bureau of Meteorology, Australia and introduce the abbreviation that you are using later in the text

P.13, L.222/230: "the reference layer" is only explained towards the end of the sub-section. It would be useful to have this part earlier in the text in order to avoid confusions. Also, please indicate which measurement device(s) are used for your calculations. Further, can you comment on / discuss the effect of snow on top of the sea ice when you calculate your heat fluxes?

P.13, L.229-233: Please rephrase & avoid repetitions of "and"

P.14, L.240-242: List references for the used constants

P.14, L.262: Again, careful with "significant"

P.15, L.277: Please explain w' and T' explicitly

P.16, L.296: check grammar / word order

P.16, L.300-303: check grammar / word order

P.16., L.312/313: Again, careful with "significant"

**Ch.4: Discussions**

P.18, L.326: How does this compare to a climatology or model results?

P.19, L.355/356: check grammar / wording

P.19, L.362: Only one sentence that relates to your own measurements? There is for sure more to discuss in the context of other studies, as well as the general context of the measurements in a large-scale and/or climatological sense.

**Ch.5: Conclusions**

P.20, L.366-371: Please rework this part (grammar, past tense, wording)

P.20, L.374: These are already the conclusions, and I still don't know how exactly snow and ice thicknesses were estimated. Please explain early in the manuscript.

P.20, L.387: "increased to twice" → doubled?

P.21, L399: "equipment malfunction" → ?? See general comment. Why is this only mentioned at the very end?

**Other aspects**

*Data availability*: "data available on request" – please consider putting your data on a public repository. Additional benefit: You'll get a proper citable DOI.

*Competing interests*: Check grammar!

*References*: L.477 & 479 – this is the same author, right? Check references for consistency.

**Figures & tables**

Fig.1:

- The photo in panel (b) is not planar as indicated on the map in panel (a), which leads to several hic-ups regarding the length-scale. Also, it seems that the distances/marked locations of the ACTD, ADV etc. are way closer together than the 30m indicated in panel (c), judging from the Ski-Doo on the right side of the photo. Please reaffirm.
- Data source and reference for the satellite image in (a) missing
- Check grammar in the caption
- Panel (c) would need a slightly better resolution (likely compression-artefacts?)

Fig.2:

- Units missing next to the colorbars
- Panel (b): Vertical gradient (btw: note the spelling mistake) → add "of temperature"
- It is not mentioned in the caption that this is a contour plot based on a limited number of measurements (four times daily)
- Please also note the year on the time axis, plus time zone (UTC? local?)
- Caption: Not mentioned how the ice surface & bottom were derived (algorithm or manually); none of the axis explained
- Colormap not suited for readers with color vision deficiencies; better examples & background for instance here: https://zenodo.org/record/5501399

Fig.3-5:

- The differentiation between 2min and 1hour average values is nowhere mentioned in the text. Either note that this is purely for visualization purposes, or justify in the text why you decided to illustrate it like that.
- Units missing in sub-panel headers (after mean/std)
- Please also note the year on the time axis, plus time zone (UTC? local?)
- In general: Anomalies in the sub-panels (b) to (i) not discussed in the paper, so either leave them out (e.g., combining the upper panels of Fig.3-5) or discuss them adequately
- Caption: Spell out / explain abbreviations

Fig.6:

- As in previous three figures: The differentiation between 40s and 10min average values is nowhere mentioned in the text. Either note that this is purely for visualization purposes, or justify in the text why you decided to illustrate it like that.
- Please also note the year on the time axis, plus time zone (UTC? local?)
- Panel (c): Add "horizontal current speed"
- Caption/panels (a) and (b): u-component / v-component
- Caption: Spell out / explain abbreviations

Fig.7:

- I would recommend to choose another symbol for "Current speed" than "s". "V" is probably more common and intuitive.
- Percentage values: why decimal values / not rounded?
- Caption: Spell out / explain abbreviations

Fig.8:

- Right y-axis: Water level anomaly?
- Left y-axis: Unit missing
- Vector-arrows: are you sure these are 2min values and not 5min?
- Caption: Spell out / explain abbreviations

Fig.9:

- Please also note the year on the time axis, plus time zone (UTC? local?)
- Please indicate the instrument from which these fluxes where derived

Fig.10:

- Caption: 2min/1hour averages, not results
- Please indicate what the error bars stand for. I assume +/- 1 standard deviation?
- As in previous figures: The differentiation between 2min and 1h average values is nowhere mentioned in the text. Either note that this is purely for visualization purposes, or justify in the text why you decided to illustrate it like that.
- Please also note the year on the time axis, plus time zone (UTC? local?)
- (b) → use different colors than in (a) and previous figures

Table 1:

- Check wording
- Add what +/- indicates

Fig.11:

- Please also note the year on the time axis, plus time zone (UTC? local?)
- Spell out abbreviations and give appropriate references
- Explain "harmonic constant calculation" (it is not in text). What exactly is merged here?!
- Are these hourly values averaged values? Then please indicate the respective standard deviations (by error bars, shading or similar).

Fig.12:

- First of all, the figure is generally hard to assess and not very intuitive. 3D plots are fancy, I know, but 2D plots might be more familiar to many potential readers.
- The caption mentions the 3D-evolutiomn of ocean velocity and direction – only, where exactly is the velocity? I see temperatures, directions, Dates (again, time zone etc. missing), salinities...but no velocities! Please explain.
- The axis-labels are generally too small

---

## Author Comment (AC1)

**Respond to Reviewer #1**

Dear reviewer,

Thank you very much for your detailed comments on the manuscript, which are constructive and will help our paper to reach a high quality. We have conducted careful revisions following your suggestions and all the comments are responded to one by one in RED.

The main updates are listed here:

(1) The time series analyzed in our manuscript has been extended from the original 10-day to7 months although several interruptions occurred during the period.

(2) The parameterizations and formulas in the revised version were further simplified, making them clear to the readers.

(3) We added a large-scale analysis that combined our observations with AMSR2 sea ice concentration and Mercator ocean products, trying to explain the long-term variations.

(4) The Results and Discussions sections were reconstructed and further analyzed.

Best regards,

The co-authors.

**A) General comments:**

**1) The presented time series of just seven days is particularly short, compared to other similar data sets in both hemispheres. How do you justify the significance of this data record? It is not 100% clear to me how this data set sets itself apart from any other. Further, you only mention an instrument malfunction at the very end of Ch.5, which is likely the cause for the short data record, right? Why is this not mentioned right at the beginning of your data description?!**

**Responses:**

Thanks for the reviewer's comments. As you mentioned, ice-ocean interface layer observations have been conducted by the previous studies, while most of the long-term time series were in the Arctic, but not in Antarctica, especially the landfast ice zone covering an entire growing season. The ocean-ice interactions in this landfast ice zone in Prydz Bay, where Chinese Zhongshan Station and Australian Davis Station were located, were affected by both local polynya and large Amery Ice Shelf, therefore our observations will help to understand this special complicated ice-glacier-ocean system.

The data interruption at the end of April was due to improper operation, which caused battery

exhaustion and observations ceased. There were two reasons we chose to analyze the short time series in the original manuscript, the first one was that we worried that interrupted data would affect the effectiveness of long-term analysis, and the second one was the ice-ocean interface layers showed the largest changes in the early frozen period, compared to other months.

However, based on the suggestions made by the reviewers, we recognized the problem of the short data in the original article, so in the revised manuscript we extended the time series to one year of observations. Although there were data interruptions in several months, the new results of the analysis are still good, and the annual variation characteristics of each element are given, which is of great significance to the study of the ice-sea interaction along the Antarctic coast.

**2) The manuscript often mentions the apparent benefits of "minute resolution" measurements, without clearly differentiating between the different data sources. For instance, the SIMBA measurements of vertical temperatures are only available four times a day (hence, imprinting on presented heat fluxes). Please revise the respective parts carefully.**

**Responses:**

Thanks for the reviewer's suggestions. In the original manuscript, the expressions were indeed misunderstood. In the revised manuscript, we cautiously use "minute resolution" and express accurately to avoid misleading readers.

In this study, the observation intervals of ADV and ACTD are 40 s and 30 s respectively, which can be used to observe the oceanic parameters on a minute scale. Based on the seawater velocity data observed by ADV and the seawater temperature and salinity data observed by ACTD, the oceanic heat flux on the 2-min scale was calculated by different parameterizations, which can reflect the instantaneous change of heat flux and capture more details of sea ice growth.

However, the SIMBA temperature chain obtains the temperature information of atmosphere-sea ice-sea water every 6 hours. Based on Stefan Law, the oceanic heat flux was calculated and analyzed. Based on these two kinds of methods, we first explored the minute-resolution oceanic heat flux in this landfast ice zone.

**3) The authors apparently decided to leave out a "traditional" chapter on the applied methodology to process and analyze the recorded data. Later in the text in the context of results (Ch.3.6), at least the heat flux calculations are explained. However, I consider these parts as misplaced. I would recommend a new separate chapter on methodical aspects & data processing prior to the results section. In this context, Chapters 3.6.1 and 3.6.2 could also be thoughtfully merged and at the same time streamlined to the most relevant aspects.**

**Responses:** Thank you for the suggestions. We moved the methods chapter (the original 3.6.1 and 3.6.2, as well as the calculation formulas of temperature and salinity) out of the Results section and formed a new Data and Methods section, which is more in line with the reading habits of the reader.

**4)    Almost all figures require a careful overhaul, be it due to the lack of proper labelling, low resolution images, non-barrier-free colormaps or "just" an insufficient / too short caption. You will find more detailed comments on all these below, right after specific comments to individual chapters.**

**Responses:** Thank you very much for your suggestion. In the revised manuscript, the pictures will be redrawn in strict accordance with your suggestion.

**5)    There is no further information on the larger scale environmental conditions (sea ice cover, atmospheric / ocean reanalysis, etc.) at all. Even if you omit to directly relate these conditions to your own data, it would be extremely helpful to have those for a proper context.**

**Responses:** According to the reviewer's suggestions, we analyzed the time series of sea ice extent, air temperature, and ocean circulation from satellite products and reanalysis datasets in the Prydz Bay in a year, and combined with the variations of ocean-sea ice heat flux calculated in this study to give a reasonable context for our study.

**6)    Please pay attention on using a (relevant) number of digits after the decimal point. Often, there is unnecessary detail given, especially when the numbers end with ".0". In addition, try to be consistent throughout the manuscript.**

**Responses:** In the revised manuscript, the appropriate reserved digits will be selected for the accuracy of the values, and the consistency of the values in the manuscript will be ensured.

**7)    You describe your results/measurements in past tense (e.g., "Figure 6 showed")  →  please use the present tense in that regard.**

**Responses:** In the revised manuscript, we will certainly pay attention to the grammar.

**B) Specific comments:**

**Abstract**

**P.1, L.14: "COMPACT-CTD"  →  there is only an ACTD mentioned in the manuscript. Please explain, also why capital letters are used here.**

**Responses:** Thanks for your reminder**.** "COMPACT-CTD" and ATCD are the same instrument. We revised to use one expression in the new version.

**P.1, L.16: Not all measurements are minute-resolution, right? SIMBA → six-hourly**

**Responses:** Yes, not all measuring instruments have a resolution of minutes, and we indeed ignored the distinctions between the time resolutions of different instruments in the abstract. Here, the sampling intervals of ACTD and ADV are 30 s and 40 s respectively, which provided the time series of ocean temperature, salinity, density, and velocity in minute resolution, while the temperature observed by SIMBA was a 6-hour interval. In the revised manuscript, strict attention will be paid to describing the sampling interval of the observation data.

**Ch.1: Introduction**

**P.2, L.37: Fraser et al. (2021) would be another good reference here (DOI: 10.5194/tc-15-5061-2021)**

**Responses:** Thanks for your suggestion and we cited it in the revised version. The deep analysis of Antarctic landfast ice in this paper makes us a deeper understanding of Antarctic landfast ice, which is worthy of my in-depth study.

**P.2, L.60: 8 psu salinity → salinity of the sea ice? More specific please**

**Responses:** Thanks for the suggestion, we changed "based on 8 psu salinity" to a clearer expression "based on a sea ice salinity of 8 psu".

**P.2, L.62: Explain abbreviation "HIGHTSI" and give the respective reference**

**Responses:** Thanks for your advice. In the revised manuscript, we explained in detail the first occurrence of abbreviations in the text and cite references appropriately as follows.

**"**High-resolution thermodynamic snow and ice model (HIGHTSI) (Launiainen and Cheng, 1998; Vihma, 2002; Cheng et al., 2006)"

Cheng, B., Vihma, T., Pirazzini, R., and Granskog, M. A.: Modelling of superimposed ice formation during the spring snowmelt period in the Baltic Sea, Ann. Glaciol., 44, 139–146, https://doi.org/10.3189/172756406781811277, 2006.

Launiainen, J. and Cheng, B.: Modelling of ice thermodynamics in natural water bodies, Cold Regions Science and Technology, 27, 153–178, https://doi.org/10.1016/S0165-232X(98)00009-3, 1998.

Vihma, T.: Surface heat budget over the Weddell Sea: Buoy results and model comparisons, J. Geophys. Res., 107, 3013, https://doi.org/10.1029/2000JC000372, 2002.

**P.3, L.67: "there are few studies" →so there are some apparently?**

**Responses:** In Prydz Bay, some previous studies calculated or simulated the oceanic heat flux by some indirect methods, but no direct observation of ocean-interface parameters. Our observations tried to establish a direct estimation of oceanic heat flux, which can fill the data gap and provide strong support for the study of landfast ice growth.

**P.3, L.75: please give a reference for the "modified Stefan's law", or explain briefly**

**Responses:** The references Zhao et al.(2019) was added to this sentence.

Zhao, J., Yang, Q., Cheng, B., Leppäranta, M., Hui, F., Xie, S., Chen, M., Yu, Y., Tian, Z., Li, M., and Zhang, L.: Spatial and temporal evolution of landfast ice near Zhongshan Station, East Antarctica, over an annual cycle in 2011/2012, Acta Oceanol. Sin., 38, 51–61, https://doi.org/10.1007/s13131-018-1339-5, 2019.

**P.3, L.77: Please explain abbreviations in the text; as they are used for the first time here**

**Responses:** Thank you for your suggestion, the second part of the article has a more specific description of the instruments used in the study, but the acronym does need to be described in detail here, which will be modified in the revised manuscript.

**P.3, L.78-80: Can be left out → phrasing in its current form**

**Responses:** The current description of the structure of the article is indeed a bit brief, which will be carefully modified in the revised manuscript so that readers can better understand the structure and content of the article.

**Ch.2: Observations**

**P.3, L.82: Coordinates misplaced; move to beginning of next sentence**

**Responses:** Revised.

**P.3, L.84: Reference for the landfast ice cover duration?**

**Responses:** We added the reference Zhao et al. (2020) in the revised version.

Zhao, J., Cheng, B., Vihma, T., Heil, P., Hui, F., Shu, Q., Zhang, L., and Yang, Q.: Fast Ice Prediction System (FIPS) for land-fast sea ice at Prydz Bay, East Antarctica: an operational service for CHINARE, Ann. Glaciol., 61, 271–283, https://doi.org/10.1017/aog.2020.46, 2020.

**P.3, L.86-88: Are these own observations or is the reference missing?**

**Responses:** Those expressions came from our previous studies, we added references here in the revised version.

**P.3, L.91-96: Please indicate reference papers/reports for the respective measurement devices (could also be moved to a table in general; together with other instrument characteristics)**

**Responses:** Revised according to the reviewer's suggestions.

**P.4, L.101: "every five days" → in a seven-day data record, it is sufficient to call that "twice"**

**Responses:** Revised.

**Ch.3: Results**

**P.5, L.120: You write that the ice-water interface was determined by a simple threshold (freezing point temperature of seawater). Can you elaborate more on that topic, especially how you handled noisy data and the given uncertainty for the IMB temperature measurements?**

**Responses:** The resolution of the SIMBA temperature sensors is 0.0625 degrees, causing noisy values appeared during observation. Therefore, 3-points smoothing is used in our data processing, which was also used in Zhao et al.(2017) and discussed in detail. They adopted a simple threshold (freezing point temperature of seawater), compared their results with the drilling observation, and found a good agreement, with the average deviation of 3.2 cm. Therefore, it is reasonable to adopt this simple threshold during the ice growth season in the winter of the southern hemisphere.

Zhao Jiechen, Yang Qinghua, Cheng Bin, et al. Snow and land-fast sea ice thickness derived from thermistor chain buoy in the Prydz Bay, Antarctic. Haiyang Xuebao, 39(11), 115-127, https://doi.org/10.3969/j.issn.0253-4193.2017.11.011, 2017.

**P.5, L.123: "observed in the field" → do you mean direct measurements, for instance by a drill?**

**Responses:** Yes, the ice thickness near the instrument are measured by winter team members at Zhongshan Station in winter by drilling.

**P.5, L.131: "2m below the ice surface" → how thick was the ice at this position?! I would assume that at least platelet ice could fairly quickly become a problem for CTD measurements at this depth…**

**Responses:** In April, at the beginning of CTD deployment, the thickness of sea ice is about 40 cm, and then the sea ice continues to thicken, about 100 cm in July and about 130 cm in September. If you look at the whole winter, the thickness of sea ice reaches its maximum in November, about 142 cm, so most of the time CTD is within the range of 50~150 cm below the ice bottom. As the reviewer said, more platelet ice has been observed at the bottom of the ice along the coast of Zhongshan Station, but we think that the CTD at the 50~150 cm below the bottom of the ice should not be affected.

**P5, L.134: with "about 0.1m above the ice bottom" – do you mean the lowest 10cm? Please rephrase**

**Responses:** Yes, the expression here may not be accurate enough, which means that the average temperature of the 10 cm at the bottom of the sea ice is -3.12±0.71°C. We changed the expressions "about 0.1 m above the ice bottom" to "the lowest 10 cm of sea ice".

**P.6, L.139/140: Please be more precise here. Which temperature gets warmer, and compared to what/when? Plus: "more heat", not "more heat flux"**

**Responses:** As mentioned in the previous analysis, there is a significant jump in ocean temperatures after April 20 compared with April 16-19, and higher ocean temperatures mean that more heat will be transferred from ocean to sea ice.

**P.7, L.147/148: Did you mix up smallest & largest deviation?**

**Responses:** Revised.

**P.8, L.158-166: Multiple remarks; Please elaborate in more detail why you used this particular equation, i.e., why you consider it as suited for your observations in Prydz Bay. Further, please do not just copy the denoted symbols from the source publication without explaining them first together with the respective units (t, S, rho). Also, be more precise and consistent with the indexing, for instance in case of salinity (ice or water salinity?).**

**Responses:** In the revised manuscript, we gave a detailed explanation of the relevant units and the words such as salinity and temperature that appeared for the first time, so that readers can understand

the parameters used in the formula more clearly.

**P.9, L.182: "ROSE analysis" is not the correct wording. It's "just" a diagram.**

**Responses:** Revised.

**P.10, L.186: Please explain what you mean by "compound current"**

**Responses:** As shown in figure 7, the observed ocean current direction was affected by topography and tide and changed with time. We tried to express this phenomenon, but used an improper word. We revised these expressions in the new version.

**P.10, L.187: Please give a proper reference / data citation for the data set from the Bureau of Meteorology, Australia and introduce the abbreviation that you are using later in the text**

**Responses:** Thanks for your suggestions, and we gave a web link reference here in the revised version.

**P.13, L.222/230: "the reference layer" is only explained towards the end of the sub-section. It would be useful to have this part earlier in the text in order to avoid confusions. Also, please indicate which measurement device(s) are used for your calculations. Further, can you comment on / discuss the effect of snow on top of the sea ice when you calculate your heat fluxes?**

**Responses:** As the reviewer's suggestion, we explained the part of the definition of "the reference layer" in an earlier position.

The snow cover doesn't affect the "the reference layer" used in our study. In the absence of vertical ice temperature observation in the previous studies, surface air temperature is often used as surface ice temperature to calculate the sea ice temperature gradient. In that condition, the snow cover will affect the calculation results. In this study, the use of vertical temperature profile data can better calculate the internal temperature gradient of sea ice and avoid the error caused by snow.

**P.14, L.240-242: List references for the used constants**

**Responses:** Thanks. We have added references to the parameters. See P.14, L.236.

**Ch.4: Discussions**

**P.18, L.326: How does this compare to a climatology or model results?**

**Responses:** We calculated the tide climatology in this region and compared with tide in our study

period. The analysis will be found in the new 4.1 section.

**P.19, L.362: Only one sentence that relates to your own measurements? There is for sure more to discuss in the context of other studies, as well as the general context of the measurements in a large-scale and/or climatological sense.**

**Responses:** We downloaded AMSR2 sea ice product and Mercator ocean product to analyze the relationship between the large scale and the small scale phenomena. The results will be shown in the discussion section.

**Ch.5: Conclusions**

**P.20, L.374: These are already the conclusions, and I still don't know how exactly snow and ice thicknesses were estimated. Please explain early in the manuscript.**

**Responses:** Thanks for the reviewer's reminds. We indeed missed the description on sea ice thickness estimations. In this study, we use simple threshold to determine the position of the ice-water interface based on SIMBA the temperature chain. Because there is no ice surface change at the observation site in winter, the upper ice surface position is fixed, and sea ice thickness can be obtained only by judging the sensor number of the ice bottom position. The explanation of SIMBA data processing will be reflected in the revised manuscript.

**P.20, L.387: "increased to twice" → doubled?**

**Responses:** Revised.

**P.21, L399: "equipment malfunction" → ?? See general comment. Why is this only mentioned at the very end?**

**Responses:** We have added the explanations at the beginning of the article. And we answered this suggestion in General comments (A).

**Other aspects**

**Data availability: "data available on request" – please consider putting your data on a public repository. Additional benefit: You'll get a proper citable DOI.**

**Responses:** Thanks for the reviewer's suggestions. We will put all the data in a public data website and get a citable DOI.

**Figures & tables**

**Fig.1:**

The photo in panel (b) is not planar as indicated on the map in panel (a), which leads to several hic-ups regarding the length-scale. Also, it seems that the distances/marked locations of the ACTD, ADV etc. are way closer together than the 30m indicated in panel (c), judging from the Ski-Doo on the right side of the photo. Please reaffirm.

Data source and reference for the satellite image in (a) missing

Check grammar in the caption

Panel (c) would need a slightly better resolution (likely compression-artefacts?)

**Responses:** Thanks for your suggestions. We revised and redraw the figure.

**Fig.2:**

Units missing next to the colorbars

Panel (b): Vertical gradient (btw: note the spelling mistake) → add "of temperature"

It is not mentioned in the caption that this is a contour plot based on a limited number of measurements (four times daily)

Please also note the year on the time axis, plus time zone (UTC? local?)

Caption: Not mentioned how the ice surface & bottom were derived (algorithm or manually); none of the axis explained

Colormap not suited for readers with color vision deficiencies; better examples & background for instance here: https://zenodo.org/record/5501399

**Responses:** Thanks for your suggestions. We revised and redraw the figure.

**Fig.3-5:**

The differentiation between 2min and 1hour average values is nowhere mentioned in the text. Either note that this is purely for visualization purposes, or justify in the text why you decided to illustrate it like that.

Units missing in sub-panel headers (after mean/std)

Please also note the year on the time axis, plus time zone (UTC? local?)

In general: Anomalies in the sub-panels (b) to (i) not discussed in the paper, so either leave them out (e.g., combining the upper panels of Fig.3-5) or discuss them adequately

Caption: Spell out / explain abbreviations

**Responses:** Thanks for your suggestions. We revised and redraw the figure.

**Fig.6:**

**As in previous three figures: The differentiation between 40s and 10min average values is nowhere mentioned in the text. Either note that this is purely for visualization purposes, or justify in the text why you decided to illustrate it like that.**

**Please also note the year on the time axis, plus time zone (UTC? local?)**

**Panel (c): Add "horizontal current speed"**

**Caption/panels (a) and (b): u-component / v-component**

**Caption: Spell out / explain abbreviations**

**Responses:** Thanks for your suggestions. We revised and redraw the figure.

**Fig.7:**

**I would recommend to choose another symbol for "Current speed" than "s". "V" is probably more common and intuitive.**

**Percentage values: why decimal values / not rounded?**

**Caption: Spell out / explain abbreviations**

**Responses:** Thanks for your suggestions. We revised and redraw the figure.

**Fig.8:**

**Right y-axis: Water level anomaly?**

**Left y-axis: Unit missing**

**Vector-arrows: are you sure these are 2min values and not 5min?**

**Caption: Spell out / explain abbreviations**

**Responses:** Thanks for your suggestions. We revised and redraw the figure.

**Fig.9:**

**Please also note the year on the time axis, plus time zone (UTC? local?)**

**Please indicate the instrument from which these fluxes where derived**

**Responses:** Thanks for your suggestions. We revised and redraw the figure.

**Fig.10:**

Caption: 2min/1hour averages, not results

Please indicate what the error bars stand for. I assume +/- 1 standard deviation?

As in previous figures: The differentiation between 2min and 1h average values is nowhere mentioned in the text. Either note that this is purely for visualization purposes, or justify in the text why you decided to illustrate it like that.

Please also note the year on the time axis, plus time zone (UTC? local?)

(b) → use different colors than in (a) and previous figures

**Responses:** Thanks for your suggestions. We revised and redraw the figure.

**Table 1:**

Check wording

Add what +/- indicates

**Responses:** Thanks for your suggestions. We revised and redraw the table.

**Fig.11:**

Please also note the year on the time axis, plus time zone (UTC? local?)

Spell out abbreviations and give appropriate references

Explain "harmonic constant calculation" (it is not in text). What exactly is merged here?!

Are these hourly values averaged values? Then please indicate the respective standard deviations (by error bars, shading or similar).

**Responses:** Thanks for your suggestions. We revised and redraw the figure.

**Fig.12:**

First of all, the figure is generally hard to assess and not very intuitive. 3D plots are fancy, I know, but 2D plots might be more familiar to many potential readers.

The caption mentions the 3D-evolutiomn of ocean velocity and direction – only, where exactly is the velocity? I see temperatures, directions, Dates (again, time zone etc. missing), salinities…but no velocities! Please explain.

**Responses:** Thanks for your suggestions. We revised and redraw the figure.

---

## Author Comment (AC2)

**Respond to Reviewer #2**

Dear reviewer,

Thank you very much for your detailed comments on the manuscript, which are constructive and will help our paper to reach a high quality. We have conducted careful revisions following your suggestions and all the comments are responded to one by one in RED.

The main updates are listed here:

(1) The time series analyzed in our manuscript has been extended from the original 10-day to 7 months although several interruptions occurred during the period.

(2) The parameterizations and formulas in the revised version were further simplified, making them clear to the readers.

(3) We added a large-scale analysis that combined our observations with AMSR2 sea ice concentration and Mercator ocean products, trying to explain the long-term variations.

(4) The Results and Discussions sections were reconstructed and further analyzed.

Best regards,

The co-authors.

**A) General comments:**

**1)   The author only gives the observation data of less than 10 days, so the representativeness of the data and whether the corresponding analysis result is robust are most worthy of discussion. If possible, the author is strongly recommended to provide longer observation data series to support the research conclusion.**

**Authors' answer:** Thanks for the reviewer's suggestions. We have extended our time series to the entire observation period, from April to December, although there were several interruptions during the period. For example, the instruments failed to work for 20 days at the end of April due to improper operation, which was caused by battery exhaustion.

We recognized the problem of the short data in the original article, so in the revised manuscript we extended the time series. Although there were data interruptions in several months, the new results of the analysis are still attractive, and the annual variation characteristics of each element are given, which is of great significance to the study of the ice-sea interaction along the Antarctic coast.

**2)   The growth and decay process of landfast sea ice is very sensitive to water depth. In this study, three observation equipment were not installed together. Although they were not far**

apart, the water depth was quite different. Therefore, how to judge the impact needs further discussion. Or it is necessary to further analyze the difference of sea ice thickness time series at the three measuring sites.

**Authors' answer:** Thanks for the reviewer's comments. When we deployed the instruments, we tried to put them in the same location, but we failed to make it because of a problem with the power system. However, this became an opportunity to see what happened to sea ice growth and sea water temperature changes when the water depth is different. The Discussion section will discuss the potential influences caused by different water depths.

3) **Limitations, errors and uncertainties of measurement and parameterization methods also need to be discussed, which are missed now.**

**Authors' answer:** Thanks for the suggestions. Indeed, many previous empirical formulas are quoted, and there are differences in different parameterization schemes. For the longer time series of oceanic heat flux calculation, we consider the equations of different parameterization schemes given by different predecessors, and it is also proved that the results are different, so we retain these results in the revised manuscript and let readers understand the differences of different equations. These analyses will be presented in the new Discussion section.

**B) Specific comments:**

**Line 28, "As a structural part of the polar ecosystem", what is the meaning of structural part here.**

**Authors' answer:** This was a writing error and is corrected in the revised version.

**Line 36, "Fast ice" use the consistent terminology pls.**

**Authors' answer:** Thank you for your suggestion. We will unify the terminology in the article and change "Fast ice" to "Landfast ice".

**Line 99, "at an accuracy of $\pm 0.0625℃$", This is the resolution, not the accuracy. Its accuracy is 0.1℃.**

**Authors' answer:** Revised.

**Line 102, "The records showed that snow and ice thickness was 0.045 m and 0.440 m on 16 April, while 0.020 m and 0.460 m" The measurement accuracy of snow and sea ice thickness**

**is 0.01m, so three decimal places are unnecessary.**

**Authors' answer:** Thank you for your comments. In the revised manuscript, we will further confirm the observation accuracy of various instruments and carefully retain the decimal places of different data sources.

**Line 115,** "**with a maximum of 4.24°C between**" **similar as Line 102. One decimal place is enough. Similar problems can be identified somewhere else.**

**Authors' answer:** Revised.

**Line 124, "However, after the 21 April, there was a decrease in the thickness of the landfast ice, with basal melt accounting for nearly 2 cm." The accuracy of SIMBA data in identifying sea ice bottom is 2 cm, so the uncertainty of melting of 2 cm here is relatively large.**

**Authors' answer:** Thanks for the comments. The sensor spacing of 2 cm limits the observation accuracy of SIMBA, we tried to smooth the data to reduce the uncertainties. The 2 cm melting at the bottom identified by SIMBA was also confirmed by the results of drilling observations in the same period, therefore we can confirm the 2 cm melting was believable.

**Line 144, "The diurnal anomalies based on the according daily mean." change to "The deviation relative to the according daily mean."**

**Authors' answer:** Revised.

**The paper has given a lot of equations, and these formulas are very basic for both ocean and sea ice physics. Therefore, I suggest that only references should be given, and it is unnecessary to list them all.**

**Authors' answer:** Thank you for your comments. In the revised manuscript, we will integrate and simplify the calculation formulas of the two methods, and give the corresponding references.

**The estimation uncertainty of ocean heat flux by the residual energy method is very dependent on the calculation time interval, and there would be large errors for high frequency calculations as shown in Figure 9.**

**Authors' answer:** Thanks for the comments. We realized that there are large uncertainties in the calculation using the residual energy method, and we compared these results with parameterization methods.

The oceanic heat fluxes calculated by the two parameterization methods also showed different

results. In the revised manuscript, we will also explore and discuss the uncertainty of the calculation results of various methods.

**Line 280, "the height of the mixing layer temperature above freezing point" change to "the deviation of the mixing layer temperature above freezing point".**

**Authors' answer:** Revised.

**Line 292, here miss the star for u as the superscript.**

**Authors' answer:** Revised.

**11, can be combined with the Fig.10.**

**Authors' answer:** Thanks. We will redraw all the illustrations in the article to complete the missing tags and use color-vision deficiency-friendly and perceptually uniform colors to make the presentation of the figures more perfect.

**The influence of tides in the study area on sea ice growth rate has been observed and analyzed, which can be referred to:**

**Lei et al., A New Apparatus for Monitoring Sea Ice Thickness Based on the Magneto strictive-Delay-Line Principle, Journal of Atmosphere and Oceanic Technology, 2009.**

**Authors' answer:** Thank you for your recommendation. We will cite this paper as a reference in the new version.

---

## Author Response (AR1)

Dear reviewer,

Thank you very much for your detailed comments on the manuscript, which are constructive and will help our paper to reach a high quality. We have conducted careful revisions following your suggestions and all the comments are responded to one by one in RED.

The main updates are listed here:

(1) The time series analyzed in our manuscript has been extended from 10 days in the original manuscript to 7 months in the revised version, although several interruptions occurred during the observation period.

(2) The manuscript was reconstructed, and especially the results and discussion sections were rewritten and further analyzed.

(3) The parameterizations and formulas in the revised version were further simplified, making them clear to the readers.

(4) We added the deep analysis between the local observations with the large-scale phenomena, like AMSR2 sea ice concentration and Mercator ocean circulation, trying to explain the seasonal variations.

Best regards,

The co-authors

**Respond to Reviewer #1**

**A) General comments:**

**1)  The presented time series of just seven days is particularly short, compared to other similar data sets in both hemispheres. How do you justify the significance of this data record? It is not 100% clear to me how this data set sets itself apart from any other. Further, you only mention an instrument malfunction at the very end of Ch.5, which is likely the cause for the short data record, right? Why is this not mentioned right at the beginning of your data description?!**

**Responses:**

Thanks for the reviewer's comments. As you mentioned, ice-ocean interface layer observations have been conducted by the previous studies, while most of the long-term time series were in the Arctic, but not in Antarctica, especially the landfast ice zone covering an entire growing season. The ocean-ice interactions in this landfast ice zone in Prydz Bay, where Chinese Zhongshan Station and Australian Davis Station were located, were affected by both local polynya and large Amery Ice

Shelf, therefore our observations will help to understand this special complicated ice-glacier-ocean system.

The data interruption at the end of April was due to improper operation, which caused battery exhaustion and observations ceased. There were two reasons we chose to analyze the short time series in the original manuscript, the first one was that we worried that interrupted data would affect the effectiveness of long-term analysis, and the second one was the ice-ocean interface layers showed the largest changes in the early frozen period, compared to other months.

However, based on the suggestions made by the reviewers, we recognized the problem of the short data in the original article, so in the revised manuscript we extended the time series to one year of observations. Although there were data interruptions in several months, the new results of the analysis were still good, and the annual variation characteristics of each element were given, which was of great significance to the study of the ice-sea interaction along the Antarctic coast.

As for the missing period of the observation series due to instrumental reasons, we have added the explanation in Section 2.1 of the revised manuscript. The corresponding changes are in lines 95–100 in the revised manuscript.

**2) The manuscript often mentions the apparent benefits of "minute resolution" measurements, without clearly differentiating between the different data sources. For instance, the SIMBA measurements of vertical temperatures are only available four times a day (hence, imprinting on presented heat fluxes). Please revise the respective parts carefully.**

**Responses:**

Thanks for the reviewer's suggestions. In the original manuscript, the expressions were indeed misunderstood. In the revised manuscript, we cautiously use "minute resolution" and express accurately to avoid misleading readers.

In this study, the observation intervals of ADV and ACTD are 40 s and 30 s respectively, which can be used to observe the oceanic parameters on a minute scale. Based on the seawater velocity data observed by ADV and the seawater temperature and salinity data observed by ACTD, the oceanic heat flux on the 2-min scale was calculated by different parameterizations, which could reflect the instantaneous change of heat flux and capture more details of sea ice growth.

However, the SIMBA temperature chain obtained the temperature information of atmosphere-sea ice-sea water every 6 hours. Based on Stefan Law, the oceanic heat flux was calculated and analyzed. Based on these two kinds of methods, we first explored the minute-resolution oceanic heat flux in this landfast ice zone.

**3)**   **The authors apparently decided to leave out a "traditional" chapter on the applied methodology to process and analyze the recorded data. Later in the text in the context of results (Ch.3.6), at least the heat flux calculations are explained. However, I consider these parts as misplaced. I would recommend a new separate chapter on methodical aspects & data processing prior to the results section. In this context, Chapters 3.6.1 and 3.6.2 could also be thoughtfully merged and at the same time streamlined to the most relevant aspects.**

**Responses:** Thank you for the suggestions. We moved the methods chapter (the original 3.6.1 and 3.6.2, as well as the calculation formulas of temperature and salinity) out of the Results section and formed a new Data and Methods section, which is more in line with the reading habits of the reader.

In the revised manuscript, the description of satellite and reanalysis products was added in section 2.2. The empirical formula and parameterizations were simplified and formed in a new section 2.3. The corresponding changes are in lines 106–162 in the revised manuscript.

**4)**   **Almost all figures require a careful overhaul, be it due to the lack of proper labelling, low resolution images, non-barrier-free colormaps or "just" an insufficient / too short caption. You will find more detailed comments on all these below, right after specific comments to individual chapters.**

**Responses:** Thank you very much for your suggestion. In the revised manuscript, the pictures were redrawn in strict accordance with your suggestion.

**5)**   **There is no further information on the larger scale environmental conditions (sea ice cover, atmospheric / ocean reanalysis, etc.) at all. Even if you omit to directly relate these conditions to your own data, it would be extremely helpful to have those for a proper context.**

**Responses:** According to the reviewer's suggestions, we analyzed the time series of sea ice extent, air temperature, and ocean circulation from satellite products and reanalysis datasets in the Prydz Bay in a year, and combined with the variations of ocean-sea ice heat flux calculated in this study to give a reasonable context for our study. The corresponding changes are in section 4.2 (lines 348–380) in the revised manuscript.

**6)**   **Please pay attention on using a (relevant) number of digits after the decimal point. Often, there is unnecessary detail given, especially when the numbers end with ".0". In addition, try to be consistent throughout the manuscript.**

**Responses:** In the revised manuscript, the appropriate reserved digits were selected for the accuracy of the values, and the consistency of the values in the manuscript was ensured.

**7) You describe your results/measurements in past tense (e.g., "Figure 6 showed") → please use the present tense in that regard.**

**Responses:** In the revised manuscript, we certainly pay attention to the grammar.

**B) Specific comments:**

**Abstract**

**P.1, L.14: "COMPACT-CTD" → there is only an ACTD mentioned in the manuscript. Please explain, also why capital letters are used here.**

Responses: Thanks for your reminder. "COMPACT-CTD" and ATCD are the same instrument. We revised to use one expression in the new version, "Conductivity–Temperature–Depth" or "CTD". The corresponding change is in line 14 of the revised manuscript.

**P.1, L.16: Not all measurements are minute-resolution, right? SIMBA → six-hourly**

**Responses:** Yes, not all measuring instruments have a resolution of minutes, and we indeed ignored the distinctions between the time resolutions of different instruments in the abstract. Here, the sampling intervals of ACTD and ADV are 30 s and 40 s respectively, which provided the time series of ocean temperature, salinity, density, and velocity in minute resolution, while the temperature observed by SIMBA was a 6-hour interval. In the revised manuscript, more attention was paid to those critical issues.

**Ch.1: Introduction**

**P.2, L.37: Fraser et al. (2021) would be another good reference here (DOI: 10.5194/tc-15-5061-2021)**

**Responses:** Thanks for your suggestion and we cited it in the revised version. The deep analysis of Antarctic landfast ice in this paper makes us a deeper understanding of Antarctic landfast ice, which is worthy of my in-depth study. The corresponding change is in line 40 of the revised manuscript.

**P.2, L.60: 8 psu salinity → salinity of the sea ice? More specific please**

**Responses:** Thanks for the suggestion, we changed "based on 8 psu salinity" to a clearer expression "based on a sea ice salinity of 8 psu".

In the revised manuscript, we re-summarized the previous research. The corresponding changes are in lines 49–62 in the revised manuscript.

**P.2, L.62: Explain abbreviation "HIGHTSI" and give the respective reference**

**Responses:** Thanks for your advice. In the revised manuscript, we explained in detail the first occurrence of abbreviations in the text and cite references appropriately as follows.

"High-resolution thermodynamic snow and ice model (HIGHTSI) (Launiainen and Cheng, 1998; Vihma, 2002; Cheng et al., 2006)". The corresponding change is in line 55 of the revised manuscript.

Cheng, B., Vihma, T., Pirazzini, R., and Granskog, M. A.: Modelling of superimposed ice formation during the spring snowmelt period in the Baltic Sea, Ann. Glaciol., 44, 139–146, https://doi.org/10.3189/172756406781811277, 2006.

Launiainen, J. and Cheng, B.: Modelling of ice thermodynamics in natural water bodies, Cold Regions Science and Technology, 27, 153–178, https://doi.org/10.1016/S0165-232X(98)00009-3, 1998.

Vihma, T.: Surface heat budget over the Weddell Sea: Buoy results and model comparisons, J. Geophys. Res., 107, 3013, https://doi.org/10.1029/2000JC000372, 2002.

**P.3, L.67: "there are few studies" ⟶so there are some apparently?**

**Responses:** In Prydz Bay, some previous studies calculated or simulated the oceanic heat flux by some indirect methods, but no direct observation of ocean-interface parameters. Our observations tried to establish a direct estimation of oceanic heat flux, which can fill the data gap and provide strong support for the study of landfast ice growth.

In the revised manuscript, we re-summarized the previous research. The corresponding changes are in lines 49–62 in the revised manuscript.

**P.3, L.75: please give a reference for the "modified Stefan's law", or explain briefly**

**Responses:** The reference Zhao et al.(2019) was added to this sentence. What is more, in the revised manuscript, we integrated the calculation methods of ocean heat flux into section 2.3. The corresponding changes are in lines 119–139 in the revised manuscript.

Zhao, J., Yang, Q., Cheng, B., Leppäranta, M., Hui, F., Xie, S., Chen, M., Yu, Y., Tian, Z., Li, M., and Zhang, L.: Spatial and temporal evolution of landfast ice near Zhongshan Station, East Antarctica, over an annual cycle in 2011/2012, Acta Oceanol. Sin., 38, 51–61, https://doi.org/10.1007/s13131-018-1339-5, 2019.

**P.3, L.77: Please explain abbreviations in the text; as they are used for the first time here**

**Responses:** Thank you for your suggestion, the second part of the article has a more specific description of the instruments used in the study, but the acronym does need to be described in detail here, which was modified in lines 68–69 in the revised manuscript.

**P.3, L.78-80: Can be left out ➙ phrasing in its current form**

**Responses:** The current description of the structure of the article is indeed a bit brief, which was carefully modified in the revised manuscript so that readers can better understand the structure and content of the article. The corresponding changes are in lines 70–73 in the revised manuscript.

**Ch.2: Observations**

**P.3, L.82: Coordinates misplaced; move to beginning of next sentence**

**Responses:** Revised.

**P.3, L.84: Reference for the landfast ice cover duration?**

**Responses:** We added the reference Zhao et al. (2020) in the revised version. The corresponding change is in line 79.

Zhao, J., Cheng, B., Vihma, T., Heil, P., Hui, F., Shu, Q., Zhang, L., and Yang, Q.: Fast Ice Prediction System (FIPS) for land-fast sea ice at Prydz Bay, East Antarctica: an operational service for CHINARE, Ann. Glaciol., 61, 271–283, https://doi.org/10.1017/aog.2020.46, 2020.

**P.3, L.86-88: Are these own observations or is the reference missing?**

**Responses:** Those expressions came from our previous studies; we added references here in the revised version. The corresponding change is in line 80.

**P.3, L.91-96: Please indicate reference papers/reports for the respective measurement devices (could also be moved to a table in general; together with other instrument characteristics)**

**Responses:** Revised according to the reviewer's suggestions.

**P.4, L.101: "every five days" ➙ in a seven-day data record, it is sufficient to call that "twice"**

**Responses:** Revised.

**Ch.3: Results**

**P.5, L.120: You write that the ice-water interface was determined by a simple threshold (freezing point temperature of seawater). Can you elaborate more on that topic, especially how you handled noisy data and the given uncertainty for the IMB temperature measurements?**

**Responses:** The resolution of the SIMBA temperature sensors is 0.0625 degrees, causing noisy values to appear during observation. Therefore, 3-point smoothing is used in our data processing, which was also used in Zhao et al.(2017) and discussed in detail. They adopted a simple threshold (freezing point temperature of seawater), compared their results with the drilling observation, and found a good agreement, with an average deviation of 3.2 cm. Therefore, it is reasonable to adopt this simple threshold during the ice growth season in the winter of the southern hemisphere.

In the revised manuscript, we used this method to process the observed data of SIMBA and combined it with the visual interpretation of manual experience to complete the extraction of the ice-ocean interface. The corresponding changes are in lines 100–101 and 171–172 in the revised manuscript.

Zhao Jiechen, Yang Qinghua, Cheng Bin, et al. Snow and land-fast sea ice thickness derived from thermistor chain buoy in the Prydz Bay, Antarctic. Haiyang Xuebao, 39(11), 115-127, https://doi.org/10.3969/j.issn.0253-4193.2017.11.011, 2017.

**P.5, L.123: "observed in the field" → do you mean direct measurements, for instance by a drill?**

**Responses:** Yes, the ice thickness near the instrument is measured by winter team members at Zhongshan Station in winter by drilling.

**P.5, L.131: "2m below the ice surface" → how thick was the ice at this position?! I would assume that at least platelet ice could fairly quickly become a problem for CTD measurements at this depth…**

**Responses:** In April, at the beginning of CTD deployment, the thickness of sea ice is about 40 cm, and then the sea ice continues to thicken, about 100 cm in July and about 130 cm in September. If you look at the whole winter, the thickness of sea ice reaches its maximum in November, about 142 cm, so most of the time CTD is within the range of 50~150 cm below the ice bottom. As the reviewer said, more platelet ice has been observed at the bottom of the ice along the coast of Zhongshan Station, but we think that the CTD at 50~150 cm below the bottom of the ice should not be affected.

Section 3.1 of the revised manuscript showed the growth of landfast ice throughout the

observation period.

**P5, L.134: with "about 0.1m above the ice bottom" – do you mean the lowest 10cm? Please rephrase**

**Responses:** Yes, the expression here may not be accurate enough, which means that the average temperature of the 10 cm at the bottom of the sea ice is -3.12±0.71℃. We changed the expression "about 0.1 m above the ice bottom" to "the lowest 10 cm of sea ice".

The corresponding change is in line 201 of the revised manuscript.

**P.6, L.139/140: Please be more precise here. Which temperature gets warmer, and compared to what/when? Plus: "more heat", not "more heat flux"**

**Responses:** As mentioned in the previous analysis, there is a significant jump in ocean temperatures after April 20 compared with April 16-19, and higher ocean temperatures mean that more heat would be transferred from the ocean to sea ice.

The corresponding change is in line 208 of the revised manuscript.

**P.7, L.147/148: Did you mix up smallest & largest deviation?**

**Responses:** Revised.

**P.8, L.158-166: Multiple remarks; Please elaborate in more detail why you used this particular equation, i.e., why you consider it as suited for your observations in Prydz Bay. Further, please do not just copy the denoted symbols from the source publication without explaining them first together with the respective units (t, S, rho). Also, be more precise and consistent with the indexing, for instance in case of salinity (ice or water salinity?).**

**Responses:** In the revised manuscript, we gave a detailed explanation of the relevant units and the words such as salinity and temperature that appeared for the first time, so that readers can understand the parameters used in the formula more clearly.

In the revised manuscript, the observations of ocean temperature, salinity and density were integrated into section 3.2, the calculation method of density was simplified, and corresponding references were given. The corresponding changes are in lines 192–229 in the revised manuscript.

**P.9, L.182: "ROSE analysis" is not the correct wording. It's "just" a diagram.**

**Responses:** Revised.

The corresponding change is in line 237 in the revised manuscript.

**P.10, L.186: Please explain what you mean by "compound current"**

**Responses:** As shown in figure 7, the observed ocean current direction was affected by topography and tide and changed with time. We tried to express this but used an improper word. We revised these expressions in the new version. The corresponding changes are in lines 231–242 in the revised manuscript.

**P.10, L.187: Please give a proper reference / data citation for the data set from the Bureau of Meteorology, Australia and introduce the abbreviation that you are using later in the text**

**Responses:** Thanks for your suggestions, and we made the corresponding revision.

**P.13, L.222/230: "the reference layer" is only explained towards the end of the sub-section. It would be useful to have this part earlier in the text in order to avoid confusions. Also, please indicate which measurement device(s) are used for your calculations. Further, can you comment on / discuss the effect of snow on top of the sea ice when you calculate your heat fluxes?**

**Responses:** As the reviewer's suggestion, we explained the part of the definition of "the reference layer" in an earlier position.

The snow cover didn't affect the "the reference layer" used in our study. In the absence of vertical ice temperature observation in the previous studies, the surface air temperature was often used as surface ice temperature to calculate the sea ice temperature gradient. In that condition, the snow cover could affect the calculation results. In this study, the use of vertical temperature profile data could better calculate the internal temperature gradient of sea ice and avoided the error caused by snow.

The corresponding changes are in lines 250–257 in the revised manuscript.

**P.14, L.240-242: List references for the used constants**

**Responses:** Thanks. We have added references to the parameters. See P.14, L.236.

**Ch.4: Discussions**

**P.18, L.326: How does this compare to a climatology or model results?**

**Responses:** We calculated the tide climatology in this region and compared it with the tide in our study period. The analysis can be found in the new 4.1 section. The corresponding changes are in lines 316–344 in the revised manuscript.

**P.19, L.362: Only one sentence that relates to your own measurements? There is for sure more to discuss in the context of other studies, as well as the general context of the measurements in a large-scale and/or climatological sense.**

**Responses:** We downloaded the AMSR2 sea ice product and Mercator Ocean product to analyze the relationship between the large-scale and the small-scale phenomena. The results were shown in the discussion section. The corresponding changes are in lines 350–367 in the revised manuscript.

**Ch.5: Conclusions**

**P.20, L.374: These are already the conclusions, and I still don't know how exactly snow and ice thicknesses were estimated. Please explain early in the manuscript.**

**Responses:** Thanks for the reviewer's reminds. We indeed missed the description of sea ice thickness estimations. In this study, we used a simple threshold to determine the position of the ice-water interface based on SIMBA the temperature chain. Because there was no ice surface change at the observation site in winter, the upper ice surface position was fixed, and sea ice thickness could be obtained only by judging the sensor number of the ice bottom position. The explanation of SIMBA data processing was reflected in the revised manuscript.

In the revised manuscript, we used 3-points smoothing to process the observed data of SIMBA and combined it with the visual interpretation of manual experience and simple threshold to complete the extraction of the ice-ocean interface. The corresponding changes are in lines 100–101 and 171–172 in the revised manuscript.

**P.20, L.387: "increased to twice" → doubled?**

**Responses:** Revised.

**P.21, L399: "equipment malfunction" → ?? See general comment. Why is this only mentioned at the very end?**

**Responses:** We have added the explanations at the beginning of the article. And we answered this suggestion in General comments (A).

**Other aspects**

**Data availability: "data available on request" – please consider putting your data on a public repository. Additional benefit: You'll get a proper citable DOI.**

**Responses:** Thanks for the reviewer's suggestions. We are considering putting all the data on a

public data website. However, we need a complex procedure now and try to make it before the manuscript is published.

**Figures & tables**

**Fig.1:**

The photo in panel (b) is not planar as indicated on the map in panel (a), which leads to several hic-ups regarding the length-scale. Also, it seems that the distances/marked locations of the ACTD, ADV etc. are way closer together than the 30m indicated in panel (c), judging from the Ski-Doo on the right side of the photo. Please reaffirm.

Data source and reference for the satellite image in (a) missing

Check grammar in the caption

Panel (c) would need a slightly better resolution (likely compression-artefacts?)

**Responses:** Thanks for your suggestions. We revised and redraw the figure.

**Fig.2:**

Units missing next to the colorbars

Panel (b): Vertical gradient (btw: note the spelling mistake) → add "of temperature"

It is not mentioned in the caption that this is a contour plot based on a limited number of measurements (four times daily)

Please also note the year on the time axis, plus time zone (UTC? local?)

Caption: Not mentioned how the ice surface & bottom were derived (algorithm or manually); none of the axis explained

Colormap not suited for readers with color vision deficiencies; better examples & background for instance here: **https://zenodo.org/record/5501399**

**Responses:** Thanks for your suggestions. We revised and redraw the figure.

**Fig.3-5:**

The differentiation between 2min and 1hour average values is nowhere mentioned in the text. Either note that this is purely for visualization purposes, or justify in the text why you decided to illustrate it like that.

Units missing in sub-panel headers (after mean/std)

Please also note the year on the time axis, plus time zone (UTC? local?)

In general: Anomalies in the sub-panels (b) to (i) not discussed in the paper, so either leave

them out (e.g., combining the upper panels of Fig.3-5) or discuss them adequately

Caption: Spell out / explain abbreviations

**Responses:** Thanks for your suggestions. We revised and redraw the figure.

**Fig.6:**

As in previous three figures: The differentiation between 40s and 10min average values is nowhere mentioned in the text. Either note that this is purely for visualization purposes, or justify in the text why you decided to illustrate it like that.

Please also note the year on the time axis, plus time zone (UTC? local?)

Panel (c): Add "horizontal current speed"

Caption/panels (a) and (b): u-component / v-component

Caption: Spell out / explain abbreviations

**Responses:** Thanks for your suggestions. We revised and redraw the figure.

**Fig.7:**

I would recommend to choose another symbol for "Current speed" than "s". "V" is probably more common and intuitive.

Percentage values: why decimal values / not rounded?

Caption: Spell out / explain abbreviations

**Responses:** Thanks for your suggestions. We revised and redraw the figure.

**Fig.8:**

Right y-axis: Water level anomaly?

Left y-axis: Unit missing

Vector-arrows: are you sure these are 2min values and not 5min?

Caption: Spell out / explain abbreviations

**Responses:** Thanks for your suggestions. We revised and redraw the figure.

**Fig.9:**

Please also note the year on the time axis, plus time zone (UTC? local?)

Please indicate the instrument from which these fluxes where derived

**Responses:** Thanks for your suggestions. We revised and redraw the figure.

**Fig.10:**

**Caption: 2min/1hour averages, not results**

**Please indicate what the error bars stand for. I assume +/- 1 standard deviation?**

**As in previous figures: The differentiation between 2min and 1h average values is nowhere mentioned in the text. Either note that this is purely for visualization purposes, or justify in the text why you decided to illustrate it like that.**

**Please also note the year on the time axis, plus time zone (UTC? local?)**

**(b) → use different colors than in (a) and previous figures**

**Responses:** Thanks for your suggestions. We revised and redraw the figure.

**Table 1:**

**Check wording**

**Add what +/- indicates**

**Responses:** Thanks for your suggestions. We revised and redraw the table.

**Fig.11:**

**Please also note the year on the time axis, plus time zone (UTC? local?)**

**Spell out abbreviations and give appropriate references**

**Explain "harmonic constant calculation" (it is not in text). What exactly is merged here?!**

**Are these hourly values averaged values? Then please indicate the respective standard deviations (by error bars, shading or similar).**

**Responses:** Thanks for your suggestions. We revised and redraw the figure.

**Fig.12:**

**First of all, the figure is generally hard to assess and not very intuitive. 3D plots are fancy, I know, but 2D plots might be more familiar to many potential readers.**

**The caption mentions the 3D-evolutiomn of ocean velocity and direction – only, where exactly is the velocity? I see temperatures, directions, Dates (again, time zone etc. missing), salinities…but no velocities! Please explain.**

**Responses:** Thanks for your suggestions. We revised and redraw the figure.

**Respond to Reviewer #2**

**A)  General comments:**

**1)  The author only gives the observation data of less than 10 days, so the representativeness of the data and whether the corresponding analysis result is robust are most worthy of discussion. If possible, the author is strongly recommended to provide longer observation data series to support the research conclusion.**

**Authors' answer:** Thanks for the reviewer's suggestions. We have extended our time series to the entire observation period, from April to December, although there were several interruptions during the period. For example, the instruments failed to work for 20 days at the end of April due to improper operation, which was caused by battery exhaustion.

We recognized the problem of the short data in the original article, so in the revised manuscript we extended the time series. Although there were data interruptions in several months, the new results of the analysis were still attractive, and the annual variation characteristics of each element were given, which was of great significance to the study of the ice-sea interaction along the Antarctic coast.

**2)  The growth and decay process of landfast sea ice is very sensitive to water depth. In this study, three observation equipment were not installed together. Although they were not far apart, the water depth was quite different. Therefore, how to judge the impact needs further discussion. Or it is necessary to further analyze the difference of sea ice thickness time series at the three measuring sites.**

**Authors' answer:** Thanks for the reviewer's comments. When we deployed the instruments, we tried to put them in the same location, but we failed to make it because of a problem with the power system. However, this became an opportunity to see what happened to sea ice growth and sea water temperature changes when the water depth was different. The potential influences caused by different water depths were in the Discussion section, in lines 369–372 in the revised manuscript.

**3)  Limitations, errors and uncertainties of measurement and parameterization methods also need to be discussed, which are missed now.**

**Authors' answer:** Thanks for the suggestions. Indeed, many previous empirical formulas were quoted, and there were differences in different parameterization schemes. For the longer time series of oceanic heat flux calculation, we considered the equations of different parameterization schemes given by different predecessors, and it was also proved that the results are different, so we retain

these results in the revised manuscript and let readers understand the differences of different equations.

The equations of different parameterization schemes were given in section 2.3, and the four results were compared in section 3.4. The large differences were mainly caused by the different formulas of friction velocity, indicating the uncertainties of the empirical equation.

**B) Specific comments:**

**Line 28, "As a structural part of the polar ecosystem", what is the meaning of structural part here.**

**Authors' answer:** This was a writing error and was corrected in the revised version. The corresponding change is in line 31 of the revised manuscript.

**Line 36, "Fast ice" use the consistent terminology pls.**

**Authors' answer:** Thank you for your suggestion. We unified the terminology in the article and change "Fast ice" to "Landfast ice".

**Line 99, "at an accuracy of $\pm0.0625℃$", This is the resolution, not the accuracy. Its accuracy is 0.1℃.**

**Authors' answer:** Revised.

**Line 102, "The records showed that snow and ice thickness was 0.045 m and 0.440 m on 16 April, while 0.020 m and 0.460 m" The measurement accuracy of snow and sea ice thickness is 0.01m, so three decimal places are unnecessary.**

**Authors' answer:** Thank you for your comments. In the revised manuscript, we confirmed the observation accuracy of various instruments and carefully retained the decimal places of different data sources.

**Line 115, "with a maximum of 4.24°C between" is similar to Line 102. One decimal place is enough. Similar problems can be identified somewhere else.**

**Authors' answer:** Revised.

**Line 124, "However, after the 21 April, there was a decrease in the thickness of the landfast ice, with basal melt accounting for nearly 2 cm." The accuracy of SIMBA data in identifying**

**sea ice bottom is 2 cm, so the uncertainty of melting of 2 cm here is relatively large.**

**Authors' answer:** Thanks for the comments. The sensor spacing of 2 cm limits the observation accuracy of SIMBA, we tried to smooth the data to reduce the uncertainties. The 2 cm melting at the bottom identified by SIMBA was also confirmed by the results of drilling observations in the same period, therefore we can confirm the 2 cm melting was believable.

**Line 144, "The diurnal anomalies based on the according daily mean." change to "The deviation relative to the according daily mean."**

**Authors' answer:** Revised.

**The paper has given a lot of equations, and these formulas are very basic for both ocean and sea ice physics. Therefore, I suggest that only references should be given, and it is unnecessary to list them all.**

**Authors' answer:** Thank you for your comments. In the revised manuscript, we integrated and simplify the calculation formulas of the two methods, and give the corresponding references.

In the revised manuscript, the description of satellite and reanalysis products is added in section 2.2. The empirical formula and parameterizations were simplified and formed in a new section 2.3. The corresponding changes are in lines 106–162 in the revised manuscript.

**The estimation uncertainty of ocean heat flux by the residual energy method is very dependent on the calculation time interval, and there would be large errors for high frequency calculations as shown in Figure 9.**

**Authors' answer:** Thanks for the comments. We realized that there are large uncertainties in the calculation using the residual energy method, and we compared these results with parameterization methods.

The oceanic heat fluxes calculated by the two parameterization methods also showed different results. In the revised manuscript, we also explored and discussed the uncertainty of the calculation results of various methods.

**Line 280, "the height of the mixing layer temperature above freezing point" change to "the deviation of the mixing layer temperature above freezing point".**

**Authors' answer:** Revised. The corresponding change is in line 150 of the revised manuscript.

**Line 292, here miss the star for u as the superscript.**

**Authors' answer:** Revised. The corresponding changes are in table 1 of the revised manuscript.

**11, can be combined with the Fig.10.**

**Authors' answer:** Thanks. We will redraw all the illustrations in the article to complete the missing tags and use color-vision deficiency-friendly and perceptually uniform colors to make the presentation of the figures more perfect.

**The influence of tides in the study area on sea ice growth rate has been observed and analyzed, which can be referred to:**

**Lei et al., A New Apparatus for Monitoring Sea Ice Thickness Based on the Magneto strictive-Delay-Line Principle, Journal of Atmosphere and Oceanic Technology, 2009.**

**Authors' answer:** Thank you for your recommendation. We cited this paper as a reference in the 4.1 section of the new version.

---

## Referee Report (RR1)

**Review on Revision #1** of (formerly) "The diurnal evolution of oceanic boundary layer beneath early-frozen landfast ice in Prydz Bay, East Antarctica"

now changed to:

**"The *annual* evolution of *ice-ocean interaction* beneath landfast ice in Prydz Bay, East Antarctica"** Hu et al. **(2022/2023)**

*January 24, 2023*
* * *
***[Please note: Not all changes in the manuscript were marked up in red, which made it unnecessarily hard to track all applied changes during review]***

The resubmitted manuscript shows some noticeable changes & inherent improvements compared to the original submission. The authors addressed many (but not all) issues that were raised by both reviewers, while in particular focusing on four core weak-points of the initial submission (time series length, MS structure, methodical section, large-scale context).

Despite all positive changes and additions, the manuscript still requires some rather comprehensive revisions. As in round one, I posted all my general and some specific comments and concerns below. Unfortunately, those again include many of my previous comments related to Figures, as most of them were not or only partially addressed in the revision of the initial manuscript (contrary to what was indicated in the author's response letter by "*We revised and redraw the figure.*"). I consider most of these comments to fall into the category of good scientific practice, but admittedly some might be more of personal preference. Still, I recommend that the authors see fit in a future revision.
* * *
**General comments**

1) As noted above, several parts of the manuscript were rearranged and changed quite noticeably. However, I feel that some of the new parts (such as Ch.4.2 or the addition of three different bulk approaches for the oceanic heat flux) are not so well connected to the rest of the manuscript. In other words – I am missing a clear and stringent "story line" on how each part of the study relates to one another. Maybe the authors can try to improve on this aspect.

2) Methods & results are again not consequently described in present tense throughout the manuscript. In addition, some numbers are still given with too many digits (e.g., heat fluxes). Please revise grammar / style.

**Specific comments *(incl. technical notes)**

**Abstract**

General: Too many brackets – please keep it concise & simple to increase readability. Plus, please check grammar/wording in order to avoid phrasing or sentences with little information.

**Ch.1: Introduction**

P.3, L.66: "in the ice-ocean model parametrization" – what exactly do you mean here? One specific parametrization or should it be a more general statement?

P.3, L.70-73: Please revise – all sentences in past tense & beginning with "the"

**Ch.2: Data and Methods**

P.3, L.83-94: Can give respective references to the individual instruments?

P.4, L.109-110: "ASI" not explained; check grammar when explaining the different temporal and spatial resolutions

P.4, L.112-113: Reference missing for the ocean reanalysis

P.4, L.114: Please be more specific about the type of interpolation and grid format.

**Ch.3: Results**

P.6, L172: "didn't change obviously" – please revise grammar

P4, L.100: You reference a detailed discussion about Ice-Ocean interface detection in „Zhao Jiechen, Yang Qinghua, Cheng Bin, et al. Snow and land-fast sea ice thickness derived from thermistor chain buoy in the Prydz Bay, Antarctic. Haiyang Xuebao, 39(11), 115-127, https://doi.org/10.3969/j.issn.0253-4193.2017.11.011, 2017." However, this paper is only available in Chinese as far as I can see. Is there also a translated version available somewhere? Otherwise, are there alternatives?

P.8, L.208/209: What do you mean with "a classic example of air – ice – ocean interactions"?

P.10, Fig.4: In addition to comments below – is this figure really necessary, given that the rose diagrams in Fig.5 already give the information on current velocities and directions? Fig.4 (panels a-c ) doesn't really yield a lot of additional information (and fluctuations are quite hard to differentiate), and a note in the text on the smaller range of W-component values could well be sufficient in my opinion.

P.12, L.270-272: Why this sudden shift to two digits?

**Ch.4: Discussions**

P.17, L.352: "Several polynyas" → Do they have names / were these already part of earlier pan-Antarctic polynya inventories? In that regard, would you expect a noticeable influence of those polynyas (e.g., by salt release through new ice production) on the oceanic measurements at Zhongshan station?

P.17, L.355-360: Is that part referring to Fig.13? If that is the case, this reference is missing. Also, this Figure is poorly introduced (what it is showing, why, etc.). Can you add a few more words on that?

P18, L368: "an obvious thick" – do you mean "thickening"?

**Other aspects**

*Data availability*: You indicated that you plan to make the data available – that's great and I would certainly see that as a strong benefit, both in terms of reproducibility of your results as well as in terms of a data sustainability.

**Figures & tables**

Several figures were noticeably modified or even exchanged completely, so additional and some of the previous comments (which still apply unfortunately!) on all figures below:

Fig.1:

- The photo in panel (b) is not planar as indicated on the map in panel (a), which leads to several hic-ups regarding the length-scale. Also, it seems that the distances/marked locations of the ACTD, ADV etc. are way closer together than the 30m indicated in panel (c), judging from the Ski-Doo on the right side of the photo. Please reaffirm.
- Indicate a reference for the Worldview-2 satellite image in (a)

Fig.2:

- Panel (b): Vertical gradient (note the spelling mistake) → add "of temperature" or use "Vertical temperature gradient"
- It is not mentioned in the caption that this is a contour plot based on a limited number of measurements (four times daily); none of the axis explained
- Please also note the year on the time axis
- Colormap not suited for readers with color vision deficiencies; better examples & background for instance here https://zenodo.org/record/5501399 or here https://tos.org/oceanography/assets/docs/29-3_thyng.pdf

Fig.3 (formerly 3-5):

- Pay attention to grammar & spelling in the caption
- Please also note the year on the time axis

Fig.4 (formerly 6):

- As in previous three figures: The reason for a differentiation between now 2 minutes and 1 hour average values is not mentioned in the text. Either note that this is purely for visualization purposes, or justify in the text why you decided to illustrate it like that.
- Please also note the year on the time axis
- Caption/panels (a) to (b): add "-component" and/or explain abbreviations

Fig.5 (formerly 7):

- I would recommend to choose another symbol for "Current speed" than "s". "V" is probably more common and intuitive.
- Caption: Too short; please be a bit more descriptive on what the sub-panels depict, on the location of measurements and the displayed quantity & unit.

Fig.6 (formerly 9):

- Indicate the reference layer / position in the caption
- It's sensible heat flux, not "specific"
- Please also note the year on the time axis

Fig.7 (formerly 10):

- Hourly and monthly mean values?
- Please indicate what the error bars stand for. I assume +/- 1 standard deviation?
- Please also note the year on the time axis
- Be a bit more descriptive in the caption – it's a bit short on information.

Table 2 (formerly 1):

- Add what +/- indicates (likely standard deviation?)

Fig.8 (formerly 11):

- Please also note the year on the time axis
- What is the temporal resolution of the displayed data?

Fig. 9:

- Please properly sort the legend in panel (b) and clearly indicate heat fluxes and current components
- Also (b): It's hard to depict any differences in the lower percentage-range. Can you try to improve this?

Fig.10:

- Caption: add unit of tidal-level bins

Fig.11:

- Unit of sea ice concentration missing (colorbar)
- Caption: reference missing; resolution of product not given
- (Small) overview map of Antarctica would help to geographically locate this area

Fig.12:

- Percentage of what? A reference area / mask (if yes, what is the spatial extent of that area?)? That is neither mentioned in the text nor indicated here (caption, sub-panel or previous Figure).
- What is the grey line with rose shading?? Again, not in caption!

Fig.13:

- Colormap not suited for readers with color vision deficiencies; better examples & background for instance here: https://zenodo.org/record/5501399
- What is "Density ocean mixed layer thickness"? I.e., what does "density" refer to (check grammar)?
- What are the white shadings? Interpolation gaps or some other data features outside the colormap range?
- Caption: reference missing; resolution of product not given

---

## Referee Report (RR2)

**Review on Revision #2** of (formerly) "The diurnal evolution of oceanic boundary layer beneath early-frozen landfast ice in Prydz Bay, East Antarctica"

now changed to:

**"Annual evolution of ice-ocean interaction beneath landfast ice in Prydz Bay, East Antarctica"**

by Hu et al. (2022/2023)

*March 31, 2023*

The resubmitted manuscript again shows some immediately noticeable changes & inherent improvements compared to the previous versions. The authors have addressed most of the comments on the clarity and completeness of Figures presented, and have also seen fit to publish the datasets quickly. Good!

In its present state, I have no further objection to its publication after a few remaining revisions. I have refrained from making a full assessment of the technical and grammatical status of the manuscript (this should take place during proofreading and technical checks of the team at TC), but all in all these corrections should be rather minor.

Similar to my last report, I have noted some remaining comments below. I would appreciate some clarification on these, but I do not need another round of review.

**General & specific comments**

-   In Ch.2.1 & 2.2: You listed weblinks as references for the individual instruments. Firstly, I would assume that the journal will require to move these to the reference section, i.e., in a more appropriate format. Secondly, as "references" I rather had published studies or reports in mind that contain/list technical specifications, data formats & examples, etc., but in case those do not exist, weblinks could be sufficient.
-   Fig.1 (caption): "False-colour satellite image (…)"
-   L.127: "Advanced Microwave Scanning Radiometer 2 (AMSR2)" (use capital letters)
-   L.339: "…which provide more details and insights for the readers and communities…" – instead of addressing different persons/groups here (I would omit this phrase), try to focus on the benefits in terms of resolvable processes that are enabled by using/showing this higher temporal resolution. In other words, more details *of what* and *compared to what* exactly?
-   L.351: "relative studies" → do you mean "related studies" here? I haven't read this wording yet, but it could certainly be correct as well.

---

## Author Response (AR2)

Dear Reviewers,

We have made further revisions to the manuscript. The comments/corrections listed by the two reviewers are taken into account and corrected accordingly.

The main revisions include the following points:
(1) We have redrawn all the figures and corrected the minor errors as the reviewers figured out. Especially we added detailed decrepitations to the figures' captions to make the figures easily understood.
(2) To benefit the ice community, the observation dataset used in this study including CTD, SIMBA, and ADV are completely released publicly and the weblinks with DOI numbers are available in the "Data availability" Section.
(3) During the revision process, one new co-author was included and listed in the fourth place. He contributed more to the results analysis and language polishing. All the other co-authors have agreed.
(4) The language has been smooth by the professional native English expert, to ensure no more grammar and spelling issues.

Please see below our response (text in red) to the reviewer's comments point by point.

Thank you for your help!

Best regards,
The authors

**Referee #1**

**General comments**

1) As noted above, several parts of the manuscript were rearranged and changed quite noticeably. However, I feel that some of the new parts (such as Ch.4.2 or the addition of three different bulk approaches for the oceanic heat flux) are not so well connected to the rest of the manuscript. In other words – I am missing a clear and stringent "story line" on how each part of the study relates to one another. Maybe the authors can try to improve on this aspect.

**Responses:** Thanks for your comments. Indeed, it was hard to find a high relationship between our local observation and the large-scale pheromone, but the sea ice evolution and the large

circulation in Prydz Bay absolutely affected the seasonal cycle of local variables. Therefore, we decided to retain the 4.2 section, but narrowed this section, removed the original figure 13 and deleted the doubtable assumptions.

To make the "storyline" clearer, we added one paragraph as the first one in the Discussion section. "The minute-frequency annual observations of variables in the ice-ocean interface in this study provide a clear picture of how they varied on an hourly, daily or seasonal scale, and fill up the data gap in Zhongshan Station. As the relative studies in other regions, those variables may be affected by the short-term cycle of sub-glacial current (McPhee et al., 1996) and ocean tide current (Lei et al., 2010). To further enrich our analysis, the relationships between processes on the local scale and pan-Prydz Bay scale were discussed here."

2) Methods & results are again not consequently described in present tense throughout the manuscript. In addition, some numbers are still given with too many digits (e.g., heat fluxes). Please revise grammar / style.

**Responses:** Thanks for your advice. In the revised manuscript, we corrected these issues.

**Specific comments** (incl. technical notes)

**Abstract**

General: Too many brackets – please keep it concise & simple to increase readability. Plus, please check grammar/wording in order to avoid phrasing or sentences with little information.

**Responses:** Thank you for your suggestion. We revised them in the new version.

**Ch.1: Introduction**

**P.3, L.66:** "in the ice-ocean model parametrization" – what exactly do you mean here? One specific parametrization or should it be a more general statement?

**Responses:** Sorry for the misleading. We revised this description in the new version.

In lines 79-81 of the new version:

"Direct observations of high-frequency ocean temperature, salinity, and velocity beneath landfast ice are important for filling the data gap of the ice–ocean interaction near the Chinese Antarctic Zhongshan Station and for more accurately understanding how the oceanic heat flux affects the growth of sea ice in Prydz Bay on the diurnal and seasonal scales."

**P.3, L.70-73:** Please revise – all sentences in past tense & beginning with "the"

**Responses:** Thanks for your advice. In the revised manuscript, we corrected these issues.

**Ch.2: Data and Methods**

**P.3, L.83-94:** Can give respective references to the individual instruments?

**Responses:** Thanks for the reviewer's suggestions. We added the references in the revised manuscript.

**In lines 99-102 of the revised manuscript:**

"A cable-type CTD sensor (model: ALEC ACTD–DF, Japanese JFE Advantech Co., Ltd.) (for more information, see

https://www.xylem.com/siteassets/brand/sontek/resources/specification/sontek-argonaut-adv-brochure-s11-02-1119.pdf, last access: February 24, 2023) was deployed 2 m beneath the ice surface and 15 m from the shoreline."

**In lines 104-107  of the revised manuscript:**

"An ADV (model: SonTek Argonaut–ADV, the xylem company) (for more information, see https://www.analyticalsolns.com.au/product/conductivity_temperature_depth_logger_miniature_ .html, last access: February 24, 2023) was deployed to observe the 3-D ocean velocity at 5 m below the ice surface and 5 m north of the CTD."

**In lines 107-110 of the revised manuscript:**

"A SIMBA (model: SRSL SIMBA) (for more information, see https://www.sams-enterprise.com/services/autonomous-ice-measurement/, last access: February 24, 2023) was deployed 5 m north of the ADV, which contained 240 temperature sensors at 2-cm intervals mounted on a thermistor string."

**P.4, L.112-113:** Reference missing for the ocean reanalysis

**Responses:** A web link was added to the revised manuscript in lines 131-135.

"The Operational Mercator global ocean reanalysis products, produced by the Copernicus-Marine Environment Monitoring Service (CMEMS), provide the daily and monthly ocean currents and mixed layer depth of the global ocean with a 1/12-degree spatial resolution and 3-hour frequency (for more information, see https://catalogue.marine.copernicus.eu/documents/QUID/CMEMS-GLO-QUID-001-030.pdf, last access: February 24, 2023)."

**P.4, L.114:** Please be more specific about the type of interpolation and grid format.

**Responses:** Thanks for the suggestion. We modified these sentences as follows:

"To facilitate comparative analysis, this study employed the nearest neighbour method to

interpolate the CMEMS products to the same projection and spatial resolution as the AMSR2 sea ice concentration."

**Ch.3: Results**

**P.6, L172:** **"**didn't change obviously" – please revise grammar

**Responses:** Thanks for your suggestion, and this sentence was modified as follows:

**In lines 196-197 of the revised manuscript:**

"The ice surface experienced no obvious changes during the cold season, therefore changes of the ice thickness mainly happened at the ice bottom."

**P4, L.100:** You reference a detailed discussion about Ice-Ocean interface detection in „Zhao Jiechen, Yang Qinghua, Cheng Bin, et al. Snow and land-fast sea ice thickness derived from thermistor chain buoy in the Prydz Bay, Antarctic. Haiyang Xuebao, 39(11), 115-127, https://doi.org/10.3969/j.issn.0253-4193.2017.11.011, 2017." However, this paper is only available in Chinese as far as I can see. Is there also a translated version available somewhere? Otherwise, are there alternatives?

Responses: Indeed, the original reference is only in Chinese and we added an English publication as a supplemental reference in the revised manuscript.

Tian Zhongxiang, Cheng Bin, Zhao Jiechen, Vihma Timo, Zhang Wenliang, Li Zhijun, Zhang Zhanhai. 2017. Observed and modelled snow and ice thickness in the Arctic Ocean with CHINARE buoy data. Acta Oceanologica Sinica, 36(8): 66–75, doi: 10.1007/s13131-017-1020-4

**P.8, L.208/209:** What do you mean with "a classic example of air–ice–ocean interactions"?

**Responses:** The original expressions are confusing and we delete this sentence in the revised manuscript.

**In lines 232-233 of the revised manuscript:**

"This event was accompanied by a concurrent increase in both the air temperature and ocean temperature, suggesting a heightened transfer of heat from both the air and ocean to the sea ice."

**P.10, Fig.4:** In addition to comments below – is this figure really necessary, given that the rose diagrams in Fig.5 already give the information on current velocities and directions? Fig.4 (panels a-c) doesn't really yield a lot of additional information (and fluctuations are quite hard to differentiate), and a note in the text on the smaller range of W-component values could well be

sufficient in my opinion.

**Responses:** Thanks for the suggestions and we deleted this figure in the revised manuscript.

**P.12, L.270-272:** Why this sudden shift to two digits?

**Responses:** Sorry for the miswriting. In the revised manuscript, we ensured the decimal format of the same variables is consistent throughout the entire text.

**Ch.4: Discussions**

**P.17, L.352:** "Several polynyas" → Do they have names / were these already part of earlier pan-Antarctic polynya inventories? In that regard, would you expect a noticeable influence of those polynyas (e.g., by salt release through new ice production) on the oceanic measurements at Zhongshan station?

**Responses:** Thanks for the reviewer's comments. We revised Figure 10 and marked these large polynyas in the new figure. The four larger polynyas are the Davis Polynya (DaP) and Four Ladies Bank Polynya (FLBP) on the east side, the Mackenzie Bay Polynya (MBP) and Cape Darnley Polynya (CDP) on the west side.

**In lines 396-399 of the revised manuscript:**

"From May to October, ice floes completely covered Prydz Bay, except for several large polynyas (Fig. 10d), for example, Davis Polynya (DaP) and the Four Ladies Bank Polynya (FLBP) on the east side and the Mackenzie Bay Polynya (MBP) and Cape Darnley Polynya (CDP) on the west side (Hou and Shi, 2021; Nihashi and Ohshima, 2015; Williams et al., 2016)."

**In lines 408-412 of the revised manuscript:**

"The four large polynyas shown in Fig. 10d started to form in April, which led to the release of a large amount of salt through new ice production during their existence. As a result, the ocean mixed layer in the corresponding locations derived from Mercator global ocean reanalysis products exhibited obvious thickening from May to October (figure not shown). In addition, the thickening of the entire ice region in Prydz Bay contributed to the strengthened vertical mixing caused by the salt rejection as the sea ice continued to grow."

[Figure]

Figure 10. (a–i) Evolution of the monthly sea ice concentration in Prydz Bay from January to December 2021. The domain of Prydz Bay (70–80°E, 65–70°S) in Antarctica is shown in the right-top corner of (a). The sea ice concentration dataset was retrieved from the AMSR2 product provided by Bremen University (https://seaice.uni-bremen.de), with a spatial resolution of 6.25 km. The locations of four large Polynyas are marked in (d), i.e., the Davis Polynya (DaP) and Four Ladies Bank Polynya (FLBP) on the east side and the Mackenzie Bay Polynya (MBP) and Cape Darnley Polynya (CDP) on the west side.

**Their references were listed as follows.**

Hou, S. and Shi, J.: Variability and Formation Mechanism of Polynyas in Eastern Prydz Bay, Antarctica, REMOTE SENSING, 13, https://doi.org/10.3390/rs13245089, 2021.

Nihashi, S. and Ohshima, K. I.: Circumpolar Mapping of Antarctic Coastal Polynyas and Landfast

Sea Ice: Relationship and Variability, JOURNAL OF CLIMATE, 28, 3650–3670, https://doi.org/10.1175/JCLI-D-14-00369.1, 2015.

Williams, G. D., Herraiz-Borreguero, L., Roquet, F., Tamura, T., Ohshima, K. I., Fukamachi, Y., Fraser, A. D., Gao, L., Chen, H., McMahon, C. R., Harcourt, R., and Hindell, M.: The suppression of Antarctic bottom water formation by melting ice shelves in Prydz Bay, Nat Commun, 7, 1–9, https://doi.org/10.1038/ncomms12577, 2016.

**P.17, L.355-360:** Is that part referring to Fig.13? If that is the case, this reference is missing. Also, this Figure is poorly introduced (what it is showing, why, etc.). Can you add a few more words on that?

**Responses:** Thanks for the comments. Considering the comments from two reviewers, we finally decided to remove Figure 13 and narrowed the relevant descriptions.

**P.18, L.362:** "an obvious thick" – do you mean "thickening"?

**Responses:** Thank you. We have revised the corresponding language problems in the revised manuscript.

**Other aspects**

**Data availability:** You indicated that you plan to make the data available – that's great and I would certainly see that as a strong benefit, both in terms of reproducibility of your results as well as in terms of a data sustainability.

**Responses:** Thanks for your comments. We realize the importance of this dataset, which is really hard to collect in the field of Antarctica, therefore we decide to share these datasets publicly with the ice community. We have uploaded them to a public data website and received the DOI number.

The observation data are available from the Science Data Bank. The seawater temperature and salinity recorded from a cable-type CTD are publicly available at https://doi.org/10.57760/sciencedb.07693 (Zhao and Hu, 2023). The air-ice-ocean temperature profile derived from Sea Ice Mass Balance Array (SIMBA) is publicly available at https://doi.org/10.57760/sciencedb.07684 (Zhao and Hu, 2023). The 3-D current velocity 5-m beneath landfast ice recorded from an Acoustic Doppler Velocimeter (ADV) are publicly available at https://doi.org/10.57760/sciencedb.07692 (Zhao and Hu, 2023).

**Figures & tables**

Several figures were noticeably modified or even exchanged completely, so additional and some of the previous comments (which still apply unfortunately!) on all figures below:

**Fig.1:**

1) The photo in panel (b) is not planar as indicated on the map in panel (a), which leads to several hic-ups regarding the length-scale. Also, it seems that the distances/marked locations of the ACTD, ADV etc. are way closer together than the 30m indicated in panel (c), judging from the Ski-Doo on the right side of the photo. Please reaffirm.

2) Indicate a reference for the Worldview-2 satellite image in (a)

**Responses:** Thanks for the suggestions. The panel (c) really confused the readers, therefore we decided to delete this subplot and retain panels (a) and (b). The distance of 30 m is a wrong estimate, and we talked to the field observer and corrected this description to "about 5 meters" in the new version. We added a web link reference for the satellite image: https://worldview.earthdata.nasa.gov.

[Figure]

Figure 1. (a) Satellite image of the observation site in Nella Fjord near Zhongshan Station, modified from the WorldView–2 multi-bands image taken on October 20 2012 (https://worldview.earthdata.nasa.gov); (b) Photo of the observation site shot down from a 30-m high slope on April 12, 2021, by Jinkai Ma, one of the co-authors. The photo is not planar as the red box in (a) because of the angle of the shot. The distances among ACTD, ADV and SIMBA were about 5 meters.

**Fig.2:**

1) Panel (b): Vertical gradient (note the spelling mistake)    add "of temperature" or use "Vertical temperature gradient"

2) It is not mentioned in the caption that this is a contour plot based on a limited number of measurements (four times daily); none of the axis explained

3) Please also note the year on the time axis

4) Colormap not suited for readers with color vision deficiencies; better examples & background for instance here https://zenodo.org/record/5501399 or here https://tos.org/oceanography/assets/docs/29- 3_thyng.pdf

**Responses:** The figure was redrawn as the reviewer suggested. The new figure and caption were shown as follows.

[Figure]

Figure 2. (a) Temperature profiles and (b) vertical gradient of the temperature profiles recorded by the SIMBA every 6 hours during April–November 2021. The white dashed line and dotted lines in (a) and (b) represent the bottom of the ice and the initial ice surface, respectively. The blue lines and red lines in (b) represent the snow surface and new ice surface after sublimation in summer.

**Fig.3 (formerly 3-5):**

1) Pay attention to grammar & spelling in the caption

2) Please also note the year on the time axis

**Responses:** The figure was redrawn and the language errors were corrected. The new figure and

caption were shown as follows.

[Figure]

Figure 3. (a) The seawater temperature observed by the CTD at 2 m beneath the landfast ice surface (blue lines), the ice temperature at the bottom (red lines; defined as the mean temperature derived by the SMIBA sensor located 0.1 m above the bottom of the ice), and air temperature observed by the SIMBA at 1 m above the landfast ice surface. (b) The seawater salinity observed by the CTD (blue lines), the seawater density calculated from the observed temperature and salinity (red lines), and the ice freezing rate at the bottom (black lines) observed by the SIMBA from April 16 to November 7.

**Fig.4 (formerly 6):**

1) As in previous three figures: The reason for a differentiation between now 2 minutes and 1 hour average values is not mentioned in the text. Either note that this is purely for visualization purposes, or justify in the text why you decided to illustrate it like that.

2) Please also note the year on the time axis

3) Caption/panels (a) to (b): add "-component" and/or explain abbreviations

**Responses:** Thanks for the suggestion. as your earlier suggestion, this figure was removed from the revised manuscript.

**Fig.5 (formerly 7):**

1) I would recommend to choose another symbol for "Current speed" than "s". "V" is probably more common and intuitive.

2) Caption: Too short; please be a bit more descriptive on what the sub-panels depict, on the location of measurements and the displayed quantity & unit.

**Responses:** Revised as the reviewer suggested. The new figure and caption were shown as follows. When we revised this figure, we found the wrong direction was displayed in the original figure. We confirm that this issue doesn't affect other results and conclusions of this study. The corrected figure was shown as follows:

[Figure]

Figure 4. Roses diagram of the horizontal current speed with a 2-minute resolution for (a) the total time series and (b-i) different months. The different colours represent the different ranges of the current speed. Due to technical issues, only 8 days were available in April and 20 days in May. Please note that the percentage scales are different in the different sub-panels.

**Fig.6 (formerly 9):**

    1) Indicate the reference layer / position in the caption

    2) It's sensible heat flux, not "specific"

    3) Please also note the year on the time axis

**Responses:** Revised the caption as the reviewer suggested. About the specific heat flux, we confirmed it in many previous studies. In this equation, we think oceanic heat flux represented the term sensible heat flux.

    The new figure and caption were shown as follows.

[Figure]

Figure 5. Conductive heat flux ($F_c$), latent heat flux ($F_l$), specific heat flux ($F_s$), and oceanic heat flux ($F_w$) were estimated using the residual method and a reference layer located 0.2 m above the bottom of the ice. The time interval is 6 hours.

**Fig.7 (formerly 10):**

    1) Hourly and monthly mean values?

    2) Please indicate what the error bars stand for. I assume +/- 1 standard deviation?

    3) Please also note the year on the time axis

    4) Be a bit more descriptive in the caption – it's a bit short on information.

**Responses:** Revised as the reviewer suggested. The new figure and caption were shown as follows.

[Figure]

Figure 7. Hourly mean $F_w$ was calculated by three bulk parameterization methods and 6-hourly mean $F_w$ was calculated by the residual method (a) and monthly mean $F_w$ (b). The error bars in (b) stand for ±1 standard deviation of hourly mean values.

**Table 2 (formerly 1):**

1)  Add what +/- indicates (likely standard deviation?)

**Responses:** Revised as follow.

"Table 2. Inter-comparisons of mean ± standard deviation of oceanic heat flux (W m$^{-2}$) calculated by different methods."

**Fig.8 (formerly 11):**

1)  Please also note the year on the time axis

2) What is the temporal resolution of the displayed data?

**Responses:** Revised as the reviewer suggested. The new figure and caption were shown as follows.

[Figure]

Figure 7. The tidal oscillations were constructed using the harmonic analysis method (Pan et al., 2018) and the harmonic constants of E et al., (2013). The temporal resolution of this dataset is 1 hour.

**Fig. 9:**

1) Please properly sort the legend in panel (b) and clearly indicate heat fluxes and current components

2) Also (b): It's hard to depict any differences in the lower percentage-range. Can you try to improve this?

**Responses:** We redraw this figure as the reviewer suggested. The new figure and caption were shown as follows.

[Figure]

Figure 8. (a) The results of the spectral analysis of the tidal oscillations and the observed ocean variables, and (b) the calculated $F_w$. The periodogram method was used to detect the periodicity (Welch, 1967).

**Fig.10:**

1) Caption: add unit of tidal-level bins

**Responses:** Revised and redraw this figure. The new figure and caption were shown as follows.

[Figure]

Figure 9. Scatter plots of the tidal level versus the oceanic variables: (a) seawater temperature, (b) seawater salinity, (c) U-component velocity, (d) V-component velocity, (e) W-component velocity, and (f–i) $F_w$ from the Bulk A, B, C, and residual methods. The grey dots are the hourly mean values of the variables, and the different lines represent the monthly mean values for 0.1 m tidal level bins.

**Fig.11:**

1) Unit of sea ice concentration missing (colorbar)
2) Caption: reference missing; resolution of product not given
3) (Small) overview map of Antarctica would help to geographically locate this area

**Responses:** Revised as the reviewer suggested and redrawn this figure. The new figure and caption were shown as follows.

[Figure]

Figure 10. (a–i) Evolution of the monthly sea ice concentration in Prydz Bay from January to December 2021. The domain of Prydz Bay (70–80°E, 65–70°S) in Antarctica is shown in the right-top corner of (a). The sea ice concentration dataset was retrieved from the AMSR2 product provided by Bremen University (https://seaice.uni-bremen.de), with a spatial resolution of 6.25 km. The locations of four large Polynyas are marked in (d), i.e., the Davis Polynya (DaP) and Four Ladies Bank Polynya (FLBP) on the east side and the Mackenzie Bay Polynya (MBP) and Cape Darnley Polynya (CDP) on the west side.

**Fig.12:**

1) Percentage of what? A reference area / mask (if yes, what is the spatial extent of that area?)? That is neither mentioned in the text nor indicated here (caption, sub-panel or previous Figure).

2) What is the grey line with rose shading?? Again, not in caption!

**Responses:** Revised as the reviewer suggested. We added a detailed description in the caption. The new figure and caption were shown as follows.

[Figure]

Figure 11. The time series of the daily percentage of open water (purple lines) relative to the domain of Prydz Bay (shown in Fig. 10) and the seawater temperature (red lines), seawater salinity (yellow lines), mean oceanic heat flux from the Bulk A, B, and C methods (grey lines with rose shading), and the oceanic heat flux from the residual method (green lines). The open water area was defined as the sum of the grid cells where the sea ice concentration was less than 15%. The rose shading indicates ±1 standard deviation.

**Fig.13:**

1) Colormap not suited for readers with color vision deficiencies; better examples & background for instance here: https://zenodo.org/record/5501399

2) What is "Density ocean mixed layer thickness"? I.e., what does "density" refer to (check grammar)?

3) What are the white shadings? Interpolation gaps or some other data features outside the colormap range?

4) Caption: reference missing; resolution of product not given

**Responses:** Thanks very much. Considering the comments from two reviewers, we narrowed this section, and removed the original figure 13, but retained the analysis text connected to this figure.

**Referee #2**

**General comments:**

1) During field deployment, how to ensure that the attitude change of ADV did not affect the

current measurement, or how to eliminate the influence during data processing?

**Responses:** Thank you for your suggestion. The ADV field observation system included the ADV probe, about 5 kg weight and two 5-m length Stainless steel cables, about 10 kg weight, therefore the ADV field system was a total of 15 kg weight. Considering the large weight, the ADV was believed to move less in terms of attitude underwater, and maintained at 5-m depth beneath the ice surface, even if it was affected by the tide and current.

However, the ADV may rotate underwater. A compass sensor was designed inside ADV to record the rotation and tilt, with the same frequency of velocity during the deployment. Furthermore, to avoid the problem of compass failure in Antarctica, we used two Stainless steel cables to deploy the ADV and kept the physical X-axis of ADV heading in the geographical East direction (90°).

Based on the records of the compass sensor and the user's settings, The program designed inside ADV automatically converts the velocity data into three geographical directions (eastward, northward, and upward). During the entire study period, we found that the largest heading changes of the physical X-axis were about 6°, accounting for less than 2% of the full circle (360°), indicating that the ADV's orientation changes were not significant and had little effect on the velocity observation.

2)  4.2 It is difficult to say that there is a direct connection between the changes in large-scale sea ice and marine environmental conditions and the local air-ice-sea interaction at the landfast ice close to the shore. Although some parameters have synchronous seasonal changes, it can only consider that the seasonal change patterns of these parameters are consistent, rather than the dynamic mechanism. In addition, it is difficult to support some of the author's assumptions in terms of the data and analysis given, thus I suggest deleting this section or only retaining some simple qualitative analysis.

**Responses:**

Thanks for the reviewer's suggestions. The discussion of large scale and local interaction was suggested by another reviewer. In fact, it was hard to find a high relationship between our local observation and the large-scale pheromone, but the sea ice evolution and the large circulation in the Prydz Bay were absolutely affected the seasonal cycle of local variables. Therefore, we decided to retained this part, but narrowed this section, removed the original figure 13 and deleted the doubtable assumptions.

**Special comments:**

1) Line 12: "The ice–ocean interaction is one of the main drivers of sea ice mass balance in the Polar Regions" --This is very vague for the study of sea ice thermodynamic process. You should emphasize the importance of high-frequency observation and high-precision estimation of ice-sea heat exchange for assessing the role of ocean heat flux in the mass balance of landfast ice.

**Responses:** Thank you for your suggestion. We revised this description as follows:

**In lines 14-15 of the revised manuscript:**

"High-frequency observations of the ice–ocean interaction and high-precision estimation of the ice–ocean heat exchange is critical to understanding the thermodynamics of the landfast ice mass balance in Antarctica."

2) The magnitude heat flux, seawater density: One decimal place is enough.

**Responses:** Thank you for your suggestion. We unified and modified the decimal places of the value in the revised manuscript. In the revised manuscript, we ensured the decimal format of the same variables is consistent throughout the entire text.

3) Line 25 "showed a typical period of 0.5 days" change to: showed a typical half-day period.

**Responses:** Thank you for your suggestion. We unified the description of values in the revised manuscript.

4) Introduction: You highlight the importance of landfast ice, and oceanic heat for ice mass balance, but miss the importance of high-frequency observation and high- precision estimation of ice-sea heat exchange. You also can give some successful cases (i.e., MOSAiC) on the estimation of oceanic heat flux under the ice based on similar observation strategy.

**Responses:** Thank you for your suggestion. We added one latest reference on sea ice mass balance and heat fluxes using dataset from MOSAiC in the revised manuscript.

"Lei, R., Cheng, B., Hoppmann, M., Zhang, F., Zuo, G., Hutchings, J. K., Lin, L., Lan, M., Wang, H., Regnery, J., Krumpen, T., Haapala, J., Rabe, B., Perovich, D. K., and Nicolaus, M.: Seasonality and timing of sea ice mass balance and heat fluxes in the Arctic transpolar drift during 2019–2020, Elementa: Science of the Anthropocene, 10, 000089, https://doi.org/10.1525/elementa.2021.000089, 2022."

5) "global warming has been becoming more and more significant,": In fact, it is not correct,

such as the hiatus of temperature rising.

**Responses:** Thank you for your suggestion. We made the necessary modifications to this portion of the information.

6) Line 76 "the second Chinese Antarctic scientific research station, which was established in February 1989 and operated year-round from then on" -- I don't think the history of Station is a useful information for your study.

**Responses:** Thank you for your suggestion. The station information can give the reader background knowledge of why we can carry out the winter observations at this station. So, we decide to retain this sentence.

7) Line 105 "when he worked as the wintering team member in Zhongshan Station": it is also useless information.

**Responses:** Thank you for your suggestion. We deleted this description in the revised manuscript.

8) Line 166 "The 4.8 m long SIMBA temperature chains recorded the vertical temperature profiles of air–snow–ice–ocean every 6 hours." moved to the section of method.

**Responses:** Thank you for your suggestion. In the revised manuscript, we have moved the corresponding description to the method section. **In lines 110-111 of the revised manuscript.**

9) Line 186 "indicating the influence of short-term weather systems on ice evolution.": The main impact should be the increase of heat content of ocean, not the increase of air temperature. The response of sea ice growth to changes in air temperature has some hysteresis.

**Responses:** Thank you for your suggestion. We incorporated your suggestion and made modifications to this part. **In lines 209-210 of the revised manuscript.**

10) Figure 2: change the sensor number to depth relative to sea level or ice surface please.

**Responses:** Revised.

11) Line 206: "which attributed to the snow isolation effect on ice and ocean." -- Cannot only attributed to the snow isolation effect.

**Responses:** Thanks for the reviewer's suggestion, and this sentence was deleted in the revised manuscript.

12) Ocean density, salinity-- would be better "seawater density and salinity".

**Responses:** Thank you for your suggestion. Compared with "ocean", "seawater" may be more appropriate to describe the observation data of the site, which has been revised in the revised manuscript.

13) Figure 5: How to explain the changes in current direction distribution and magnitude from April to May?

**Responses:** The reason for the significant changes from April to May in Figure 5 was the length of the available data, only 8 days in April and 20 days in May, due to technical issues. We explained it in the caption of Figure 4 in the revised manuscript.

"Figure 4. Roses diagram of the horizontal current speed with a 2-minute resolution for (a) the total time series and (b-i) different months. The different colours represent the different ranges of the current speed. Due to technical issues, only 8 days were available in April and 20 days in May. Please note that the percentage scales are different in the different sub-panels.
"

14) Line 308 "methods were consistent with the previous studies but based on a higher temporal resolution": It is difficult to say that the difference is tens of watts is "consistent". It can only be said that the seasonality of oceanic heat flux given by different methods is consistent. The quantitative difference may be related to specific methods and environmental parameters of the given year. In addition, the estimation of oceanic heat flux at the ice bottom based on the residual energy method will produce great errors in a short time window.

**Responses:** Thank you for your suggestion. We restructured this part as follows in the revised manuscript.

**In lines 336-341 of the revised manuscript:**

"In this study, the average oceanic heat flux calculated using the residual method and the bulk methods are consistent with those of previous studies on the seasonal scale, and the quantitative difference may be related to the specific methods and environmental parameters for the given years. In this study, we utilized a higher temporal resolution (6 hours for the residual method and 2 minutes for the bulk methods), which provide more details and insights for the readers and communities, while the estimation of the oceanic heat flux at the bottom of the ice based on the residual method may produce great errors within a short time window (Lei et al., 2010)."

15) "E et al., (2013).": It is from Huang et al. (2013). Right?

**Responses:** This reference is E et al., (2013), the first author is E Dongchen, from Wuhan University, China.

16) "PSD peaks": what is the PSD here?

**Responses:** "PSD (power spectral density)", we have made this part clearer in the revised manuscript.

**In lines 357-359 of the revised manuscript:**

"In this study, the periodogram method (Welch, 1967) was used to detect the periodicity of the long time-series observation data. Power spectrum analysis of the signal revealed that the tidal oscillations exhibited two peaks."

17) Figure 9: "The results of spectral analysis": the result is very strange for me? How to obtain such probability distribution?

**Responses:** Thank you for your comments. The method used for the results in Figure 8 is detailed in the previous reply.

This study used the Periodogram method to detect the periodicity of long time-series observation data. The Periodogram is a method of estimating the Power Spectral Density (PSD) of a signal and is primarily used to estimate the power distribution of a signal at different frequencies to detect whether there is a clear periodicity in the signal. If a signal has periodicity, then the power at certain specific frequencies will be much higher than at other frequencies. These frequencies are the signal's periodic frequencies, and their power density reflects the periodicity of the signal.

---

## Author Response (AR3)

Dear Reviewers,

We have made further revisions to the manuscript. The comments/corrections listed by the two reviewers are considered and corrected accordingly. Please see below our response (text in red) to the reviewer's comments point by point.

Thank you for your help!

Best regards,
The authors

**Review on Revision #1**

**Special comments**

1) Line 31 "long-term observations" change to "Cross-seasonal observations".

**Responses:** Thanks for your advice. **In line 34 of the revised manuscript,** we corrected it in the revised manuscript.

2) Line 34-40: I don't think the climate and sea ice changes in the Arctic have anything to do with your study, as I said last time.

**Responses:** Thanks for your advice. In the revised manuscript, we removed these descriptions.

3) Line 90 "the second Chinese Antarctic scientific research station, which was established in February 1989 and has been operated year-round since its establishment" -- just delete it.

**Responses:** Thanks for your advice. We have deleted these descriptions in the revised manuscript.

4) The Operational Mercator global ocean reanalysis products: I do not believe that the reanalysis data of the ocean has the ability to solve the ocean processes under landfast ice with only <100 m from the shore, that having complex topographic features.

**Responses:** Yes, we agreed with the reviewer's opinion. In fact, the Mercator ocean reanalysis was used to map the large-scale distribution of ocean temperature and current, not the small-scale. We made it clear in the revised manuscript as shown in Section 2.2.

5) Line 199 and other texts: "The annual mean": Actually, you don't have a year of observational data.

**Responses:** Thanks for your advice. We corrected it in the revised manuscript.

6) Line 202 "which are similar to previous observations", Further comparison with the following study is necessary because the landfast ice near Zhongshan Station will be affected by the terrain and distribution of grounding icebergs.

Lin L, Lei R, Hoppmann M, Perovich D K, and He H. 2022. Changes in the annual sea ice freeze–thaw cycle in the Arctic Ocean from 2001 to 2018, The Cryosphere, 16, 4779–4796, https://doi.org/10.5194/tc-16-4779-2022, 2022.

**Responses:** Thanks for your advice. **In lines 200−202 of the revised manuscript,** we cited the related paper and revised it as follows:

"which are similar to the nearshore observations at Zhongshan Station in 2006 (Lei et al., 2010) and in 2012 (Zhao et al., 2019), but different to the offshore cases around this region, especially when grounded icebergs existed (Li et al., 2023) ."

The new reference is listed as follow:

Li, N., Lei, R., Heil, P., Cheng, B., Ding, M., Tian, Z., and Li, B.: Seasonal and interannual variability of the landfast ice mass balance between 2009 and 2018 in Prydz Bay, East Antarctica, The Cryosphere, 17, 917–937, https://doi.org/10.5194/tc-17-917-2023, 2023.

7) Line 210 "indicating the influence of the air–ice–ocean interactions on the ice evolution": just delete it, it is a correct but meaningless discussion.

**Responses:** Thanks for your advice. We have modified the corresponding description in the revised manuscript.

8) Figure 2: It is best to provide both snow and ice thickness in-situ observation data. In addition, I do not believe that during some winter periods, there were no snow at all, and there was still 2-6 cm of snow.

**Responses:** Thanks for your advice. Unfortunately, the in-situ observations of snow and ice thickness were not available for us at this moment. According to the previous studies, the snow and ice thickness retrieved from SIMBA were reliable to demonstrate the snow and ice annual evolution (Zhao et al., 2021; Lei et al., 2022).

The new references are listed as follow:

Zhao Jiechen, Yang Tao, Shu Qi, Shen Hui, Tian Zhongxiang, Hao Guanghua, Zhao Biao. 2021. Modelling the annual cycle of landfast ice near Zhongshan Station, East Antarctica. Acta Oceanologica Sinica, 40(7): 129–141, doi: 10.1007/s13131-021-1727-0

Lei, R, Cheng, B, Hoppmann, M, Zhang, F, Zuo, G, Hutchings, JK, Lin, L, Lan, M, Wang, H, Regnery, J, Krumpen,T, Haapala, J, Rabe, B, Perovich, DK, Nicolaus, M. 2022. Seasonality and timing of sea ice mass balance and heat fluxes in the Arctic transpolar drift during 2019–2020. Elementa: Science of the Anthropocene 10(1). DOI: https://doi.org/10.1525/ elementa.2021.000089

9)  "after sublimation in summer": How do you tell if it's sublimation rather than melting.

**Responses:** Thanks for your advice. **In lines 215−216 of the revised manuscript,** we corrected it to "sublimation or melting in summer".

10) Figure 3: The density of seawater does not make much sense, and the deviation of seawater temperature above freezing point should be given.

**Responses:** Thanks for your advice. We corrected these problems in the revised manuscript, and the figure was redrawn. The new figure and caption were shown as follows.

[Figure]

Figure 3. (a) The seawater temperature observed by the CTD at 2 m beneath the landfast ice surface (blue lines),

the ice temperature at the bottom (red lines; defined as the mean temperature derived by the SMIBA sensor located 0.1 m above the bottom of the ice), and air temperature observed by the SIMBA at 1 m above the landfast ice surface. (b) The seawater salinity observed by the CTD (blue lines), the deviation of seawater temperature above freezing point (ΔT, red lines), and the ice freezing rate at the bottom (black lines) observed by the SIMBA from April 16 to November 7.

**In lines 257−262 of the revised manuscript:**

"After acquiring the seawater salinity by CTD, the seawater freezing point was calculated with the observed seawater temperature and salinity, using the equation proposed by Millero (1978). The calculated freezing point decreases with the increase of the seawater salinity, from −1.83℃ in April to −1.86℃ in May, and then remained stable, with a mean value of −1.87℃ in the following seasons. Further, the deviation of seawater temperature above the freezing point was calculated (ΔT, red lines in Fig. 3b), which increased quickly from 0.15°C to 0.55°C in April and decreased to around 0.1°C in the middle of May and maintained to November."

11) Line 252-255 just remove it.

**Responses:** Thanks for your advice. In the revised manuscript, we redraw the figure and remove the lines of density, while we retain the text in the revised version.

12) Line 265 "the horizontal current exhibited a similar distribution in all three directions": there are three directions for the horizontal current?

**Responses:** Sorry for the spelling mistake. We revised it in the new version.

**In line 273 of the new version:** "the horizontal current exhibited a similar distribution in the directions".

13) Figure 4 "Please note that the percentage scales are different in the different sub-panels.": use the same scale please.

**Responses:** Revised as the reviewer suggested. The new figure and caption were shown as follows.

[Figure]

Figure 4. Roses diagram of the horizontal current speed with a 2-minute resolution for (a) the total time series and (b–i) different months. The different colours represent the different ranges of the current speed. Due to technical issues, only 8 days were available in April and 20 days in May.

14) Line 285 "but a practical value is more realistic in the Fc": it is in the Fl?

**Responses:** Sorry for the misleading. We revised this description in the new version.

**In line 292 of the new version:** "but a practical value is more realistic in the $F_l$ calculation".

When we checked the Fig. 5, we found that the notation of $F_l$, $F_c$ and $F_s$ in the original figure was inverted when drawing the figure, but it will not affect the final oceanic heat flux calculation result. **In lines 297−304 of the new version,** the description errors only appear in this paragraph and have been corrected. We confirm that this issue doesn't affect other results and conclusions of this study. The corrected figure was shown as follows:

[Figure]

Figure 5. Conductive heat flux ($F_c$), latent heat flux ($F_l$), specific heat flux ($F_s$), and oceanic heat flux ($F_w$) were estimated using the residual method and a reference layer located 0.2 m above the bottom of the ice. The time interval is 6 hours.

15) Table: remove it because it give the information repeated from the above figure.

**Responses:** Thanks for your advice. In the revised manuscript, we removed this Table.

16) Line 351 "and fill up the data gap" -- change to "fill up the knowledge gap"

**Responses:** Thanks for your advice. **In line 357 of the revised manuscript,** we revised this description in the new version.

17) Line 487 "such as ice thickness radar, ocean temperature chains, and ice salinity gauges": Actually, I don't know what are the ice thickness radar and ice salinity gauges.

**Responses:** Thanks for your advice. **In line 493 of the revised manuscript,** we revised this description in the new version.

**Referee #2**

**General & specific comments:**

1) In Ch.2.1 & 2.2: You listed weblinks as references for the individual instruments. Firstly, I would assume that the journal will require to move these to the reference section, i.e., in a more appropriate format. Secondly, as "references" I rather had published studies or reports in mind that contain/list technical specifications, data formats & examples, etc., but in case those do not exist, weblinks could be sufficient.

**Responses:** Thank you for your suggestion. Unfortunately, we have not been able to find studies or reports that include/list technical specifications, data formats, examples, etc., so the web link is listed directly here as a reference to the instrument parameters.

2) Fig.1 (caption): "False-colour satellite image (…)"

**Responses:** Thanks for your advice. **In line 121 of the revised manuscript,** we revised this description in the new version.

3) L.127: "Advanced Microwave Scanning Radiometer 2 (AMSR2)" (use capital letters)

**Responses:** Thanks for your advice. **In lines 126−127 of the revised manuscript,** we revised this description in the new version.

4) L.339: "…which provide more details and insights for the readers and communities…" – instead of addressing different persons/groups here (I would omit this phrase), try to focus on the benefits in terms of resolvable processes that are enabled by using/showing this higher temporal resolution. In other words, more details of what and compared to what exactly?

**Responses:** Thanks for your advice. We revised this description in the new version.

**In lines 345−349 of the new version:** "Compared to the higher temporal resolution (6 hours for the residual method and 2 minutes for the bulk methods) in this study, the estimation based on the traditional borehole observations may produce great errors within a short time window (Lei et al., 2010). Therefore, this high-frequency observation can more accurately capture the subtle changes of oceanic heat flux in the short term, and better analyze the annual evolution of the ice–ocean interaction.".

5) L.351: "relative studies" → do you mean "related studies" here? I haven't read this wording yet, but it could certainly be correct as well.

**Responses:** Thanks for your advice. **In line 357 of the revised manuscript,** we revised this description in the new version.

---

## Author Response (AR4)

Dear Editor,

We are very glad to receive your email. Thank you for your recognition of this article and your help all the time. In view of the two minor technical revisions you proposed, we have made corresponding modifications in the revised manuscript. Please see below our response (text in red) to the comments point by point.

Thank you for your help!

Best regards,
The authors

**Technical revisions**

1. Please do indeed move the links to the technical descriptions into the references, and just cite them properly, e.g. as: (Manufacturer, Year)

**Responses:** Thanks for your advice. **In lines 99–104 of the revised manuscript,** we modified the format of the citations and added technical descriptions to the references.

2. Please change SEE and SWW to ESE and WSW, I think that is the more appropriate abbreviation if I understand correctly what you mean?

**Responses:** Thanks for your advice. In the revised draft, we have changed SEE and SWW to ESE and WSW.